# When Silence Is Golden: Can LLMs Learn to Abstain in Temporal QA and Beyond?

**Xinyu Zhou**[*]
HKUST (GZ)

**Chang Jin**[*]
Tongji University

**Carsten Eickhoff**
University of Tübingen

**Zhijiang Guo**
HKUST, HKUST (GZ)

**Seyed Ali Bahrainian**
University of Tübingen

## ABSTRACT

Large language models (LLMs) rarely admit uncertainty, often producing fluent but misleading answers, rather than abstaining (i.e., refusing to answer). This weakness is even evident in temporal question answering, where models frequently ignore time-sensitive evidence and conflate facts across different time-periods. In this paper, we present the first empirical study of training LLMs with an abstention ability while reasoning about temporal QA. Existing approaches such as calibration might be unreliable in capturing uncertainty in complex reasoning. We instead frame abstention as a teachable skill and introduce pipelines including one that couples Chain-of-Thought (CoT) supervision with Reinforcement Learning (RL) guided by abstention-aware rewards. Our goal is to systematically analyze how different information types and training techniques affect temporal reasoning with abstention behavior in LLMs. Through extensive experiments studying various methods, we find RL yields strong gains on reasoning: a model initialized by Qwen2.5-1.5B-Instruct surpasses GPT-4o by $3.46\%$ and $5.80\%$ in Exact Match on TimeQA-Easy and -Hard, respectively. Moreover, it improves the True Positive rate on unanswerable questions by $20\%$ over a pure supervised fine-tuned (SFT) variant. Beyond performance, our analysis shows that SFT induces overconfidence and harms reliability, while RL improves prediction accuracy but exhibits similar risks. Finally, by comparing implicit reasoning cues (e.g., original context, temporal sub-context, knowledge graphs) with explicit CoT supervision, we find that implicit information provides limited benefit for reasoning with abstention. Our study presents new insights into how abstention and reasoning can be jointly optimized, providing a foundation for building more reliable LLMs. Dataset and code is publicly released `https://github.com/Blackzxy/AbstentionTemporalQA`.

## 1 INTRODUCTION

Large Language Models (LLMs) have demonstrated impressive performance across a variety of question answering (QA) tasks (Raffel et al., 2020; Ouyang et al., 2022b). However, they frequently generate confident yet incorrect responses, even when lacking sufficient knowledge or when the question is inherently unanswerable (Kirichenko et al., 2025; Tomani et al., 2024). This tendency to hallucinate answers can mislead users, compromise model reliability, and pose serious risks in high-stakes domains such as medicine (Asgari et al., 2025) and law (Dahl et al., 2024). To assess this limitation, we evaluate GPT-4o on the unanswerable part of the TimeQA dataset (Chen et al., 2021), which is used to test the LLMs' reasoning and abstention ability, where "unanswerable" denotes questions that do not have a correct answer due to contradictory or ambiguous information in the context. As illustrated in Figure 1, GPT-4o performs poorly in the unanswerable subset, highlighting the challenges that LLMs face in reliable abstention.

Among the various QA tasks, those involving temporal information present even greater challenges. Temporal QA requires models to reason about evolving events, shifting timelines, and often ambiguous temporal expressions (Wei et al., 2025; Wallat et al., 2025). Unlike static, fact-based queries,

---

[*]Equal Contributions.

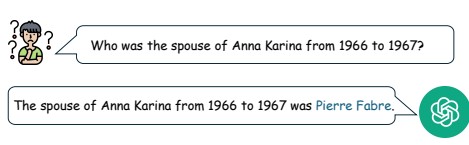

(a) An example of how GPT-4o fails to correctly abstain from answering a time-related question when there is no exact answer.

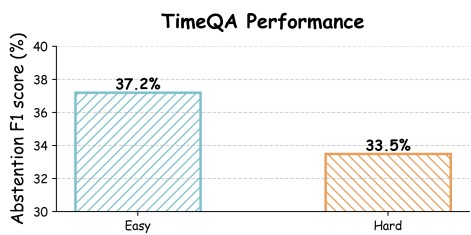

(b) GPT-4o's zero-shot F1 score on the unanswerable part of TimeQA, with context provided.

Figure 1: Qualitative and quantitative illustrations of LLM's abstention ability. Both demonstrate that temporal abstention is still challenging for LLMs.

temporal questions demand a nuanced understanding of how events unfold over time and how their temporal relationships impact the question's answerability (Islakoglu & Kalo, 2025). This complexity is further heightened when models are expected not only to reason temporally but also to abstain in cases of insufficient or contradictory temporal information. Importantly, abstention is especially common and necessary in temporal question–answering scenarios: users frequently ask about events that are dynamic such that the facts and correct answers change over time, rendering the facts outdated. In such cases, models must decide whether the question is answerable, making temporal QA a natural real-world setting where abstention behavior is both needed and essential. As such, temporal QA with refusal capability serves as a rigorous test of LLM reliability (Chen et al., 2023; Wen et al., 2025). For instance, as illustrated in Figure 1a, when asked "Who was the spouse of Anna Karina from 1966 to 1967?", the model incorrectly responds "Pierre Fabre", failing to account for their divorce in 1965. This example underlines the model's inability to identify unanswerable, time-sensitive queries and its tendency to generate confident but erroneous answers.

Efforts to improve LLMs' refusal capabilities span calibration (Jiang et al., 2021; Press et al., 2022; Tian et al., 2023), training adaptations (Azaria & Mitchell, 2023; Cobbe et al., 2021; Kadavath et al., 2022; Ouyang et al., 2022c; Slobodkin et al., 2023), prompting (Feng et al., 2023; Xu et al., 2023), consistency-based reasoning (Ding et al., 2023a; Wang et al., 2022), and multi-model aggregation (Feng et al., 2024b). While partially effective, these methods have limitations: calibration misrepresents uncertainty in complex reasoning, and training-based approaches often require additional supervision, limiting generalization (Feng et al., 2024b). Although these approaches have achieved encouraging results on standard QA tasks, their effectiveness in temporally sensitive QA remains largely unexplored. The ambiguity in temporal expressions (e.g., phrases like "before" and "after") and contradictions (e.g., the example shown in Figure 1) in temporal information make abstention decisions more difficult, leading to significantly worse performance in this setting, as illustrated in Figure 1. This aspect received little attention in prior work, leaving a gap in the literature.

Our primary goal is not merely to develop a stronger model, but to deeply understand how different reasoning cues (e.g., original context, sub-context related to the timestamp in the question, and knowledge graphs extracted from the original context), model sizes, and various training strategies affect LLMs' temporal reasoning with abstention. To this end, we investigate how LLMs perform in various settings (e.g., different scales, input information types, etc.) on temporally grounded questions that may lack definitive answers. To validate whether RL can enhance LLMs' temporal reasoning with abstention abilities, we also propose a pipeline integrated with CoT (Wei et al., 2022) supervision and RL, with a carefully designed reward to encourage honesty and reduce hallucinations. Surprisingly, the RL-tuning model with only 1.5B parameters can achieve $5.80\%$ improvement in Exact-Match than GPT-4o on TimeQA-Hard. We highlight that smaller models benefit more from the CoT-SFT cold start than knowledge graphs or contexts, which is essential to elicit reasoning ability. Through an exhaustive analysis, we identify a trade-off between overall accuracy and abstention ability, while designing effective rewards and determining the optimal training data ratio remain challenging for robust abstention.

## 2   RELATED WORK

Recent work has explored enabling LLMs to abstain from answering when uncertain, using a variety of strategies. Calibration-based methods adjust model confidence to better reflect correctness.

For example, Press et al. (2022) propose elicitive prompting (e.g., CoT, self-ask) to address compositionality gaps, while Tian et al. (2023) show that RLHF-tuned models produce more reliable verbalized confidence. Training-based approaches incorporate auxiliary objectives or modules to help models identify unanswerable inputs (Azaria & Mitchell, 2023; Cobbe et al., 2021; Slobodkin et al., 2023). Prompt-based methods enhance reasoning transparency through self-asking or context optimization (Feng et al., 2023; Xu et al., 2023). Consistency-based approaches assess uncertainty by measuring agreement across outputs (Wang et al., 2022; Ding et al., 2023a), while multi-LLM methods improve robustness by aggregating predictions and abstaining in the absence of consensus (Feng et al., 2024b). Song et al. (2025) has recently explored the LLM's refusal behaviors in the math domain using RL techniques. Our work is the first to systematically study abstention in temporally grounded QA, an area where dynamic event structures and ambiguous temporal expressions make reliable abstention particularly challenging. We identify key failure modes and propose an effective strategy that improves abstention performance in this complex setting. We provide the extended related work in Appendix D.

## 3 PRELIMINARIES

### 3.1 PROBLEM DEFINITION OF TEMPORAL QA WITH ABSTENTION

In this study, we focus on the task of question answering with abstention, which requires LLMs to abstain based on the internal knowledge (Feng et al., 2024a; Whitehead et al., 2022) or the external context if available. In particular, we further concentrate on a more challenging variant, as shown in Figure 1: the temporal abstention question-answering task, which involves time-sensitive questions to assess the temporal reasoning ability of LLMs.

We consider a temporal QA dataset containing $\mathcal{D}_{\texttt{train}} = \{(q_i, c_i, a_i)\}_{i=1}^N$ and $\mathcal{D}_{\texttt{test}} = \{(q_j, c_j, a_j)\}_{j=1}^M$, where $q_i, c_i, a_i$ denote the question, the question-relevant context, and the ground-truth answer. In the abstaining QA setting, the language model (LM) $\pi_{\boldsymbol{\theta}}$ with parameters $\boldsymbol{\theta}$ is required to generate the correct answer if it exists, or abstain (e.g., generate "No Answer") if the question is unanswerable. Each question $q_i$ is accompanied by auxiliary information $e_i$, which may consist of the original context $c_i$, or other derived evidence (e.g., Knowledge Graphs, KG) extracted from $c_i$. Formally, we define the model's output $o_i$ for a question $q_i$ as: $o_i = \pi_{\boldsymbol{\theta}}(q_i, e_i)$.

### 3.2 GROUP RELATIVE POLICY OPTIMIZATION (GRPO)

One method we study is based on GRPO (Shao et al., 2024; DeepSeek-AI et al., 2025), a popular RL approach that has recently shown great promise in aligning the behavior of LLMs with a given signal. We aim to extend this approach to the abstention task. GRPO first samples a group of outputs $\{o_i^{(1)}, o_i^{(2)}, ..., o_i^{(G)}\}$ on the $i$-th question $q_i$ from the policy model $\pi_{\boldsymbol{\theta}}$, then optimizes it by maximizing the following objective:

$$\mathcal{J}_{\texttt{GRPO}} = \mathbb{E}_{[\boldsymbol{q} \sim \mathcal{D}, \{\boldsymbol{o}^{(i)}\}_{i=1} \sim \pi_{\boldsymbol{\theta}}(\cdot|\boldsymbol{q})]}$$
$$\left[ \frac{1}{G} \sum_{i=1}^G \left( \min(\frac{\pi_{\boldsymbol{\theta}}(\boldsymbol{o}^{(i)}|\boldsymbol{q})}{\pi_{\boldsymbol{\theta}_{old}}(\boldsymbol{o}^{(i)}|\boldsymbol{q})} A_i, \texttt{clip}(\frac{\pi_{\boldsymbol{\theta}}(\boldsymbol{o}^{(i)}|\boldsymbol{q})}{\pi_{\boldsymbol{\theta}_{old}}(\boldsymbol{o}^{(i)}|\boldsymbol{q})}, 1 - \epsilon, 1 + \epsilon) A_i) - \beta \mathbb{D}_{\texttt{KL}}(\pi_{\boldsymbol{\theta}} || \pi_{ref}) \right) \right]$$
$$(1)$$

where $\beta$ is a hyper-parameter, $A_i$ denotes the advantage, $\boldsymbol{r} = \{r_1, r_2, ..., r_G\}$ denotes the corresponding rewards given a group of outputs $\{o_i^{(1)}, o_i^{(2)}, ..., o_i^{(G)}\}$. $\mathbb{D}_{\texttt{KL}}$ is an unbiased estimator of KL divergence (Schulman, 2020), to penalize a new policy from being too far from the reference model. The detailed definition can be found in Appendix E.

## 4 METHODS

In this section, we first introduce several reasoning information extraction variants designed for this task for subsequent SFT and RL training. We categorize the extraction methods into two types: (1) implicit reasoning information extraction, which provides important background context and knowledge without explicitly guiding reasoning steps; (2) explicit reasoning information extraction, which exposes step-by-step reasoning through CoT traces. Given that the context $\boldsymbol{c}$ is available in the TimeQA dataset (Chen et al., 2021), we focus on applying the information extraction variants

on $c$. We then present our reinforcement learning framework, which utilizes the extracted reasoning information to improve both reasoning quality and abstention behavior.

## 4.1 IMPLICIT REASONING INFORMATION EXTRACTION

We regard contextual signals that indirectly aid reasoning as *implicit reasoning information*. In our framework, we focus on two types: (1) **time-related information**, which filters temporally relevant sub-contexts, and (2) **knowledge graph (KG) facts**, which provide structured temporal knowledge. The extracted implicit reasoning information will be concatenated with the original question and provided as input to the LLM.

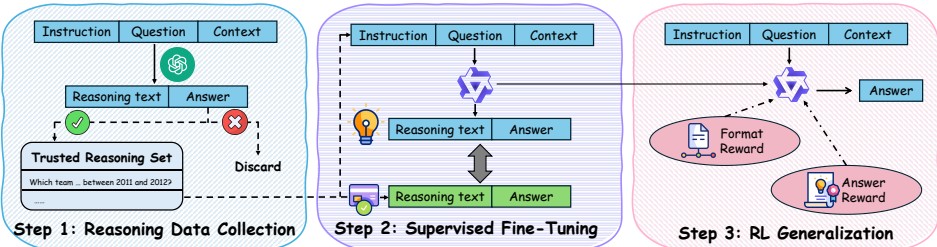

Figure 2: Overview of the CoT+RL pipeline. High-quality CoT reasoning data are first generated and filtered to form a trusted reasoning set, which is used for SFT. The SFT model is then further optimized with RL to enhance reasoning and abstention capabilities using format and answer reward.

### 4.1.1 TIME-RELATED INFORMATION EXTRACTION

Given a question $q$ associated with timestamp $t_q$ and background context $c$, we extract a time-related sub-context $tc \subset c$ containing only facts relevant to $t_q$. Formally:

$$tc = \text{ExtractTimeRelated}(q, c),$$

where the extraction is performed via GPT-4o-mini[1] with a time-filtering prompt. This reduces noise from irrelevant temporal facts. Full prompt is provided in subsection H.1. The detailed time-related information extraction pipeline is presented in subsection F.1.

### 4.1.2 KG EXTRACTION

To enhance reasoning with structured knowledge, we retrieve temporal KG facts represented as quadruples $(h, r, t, \tau)$, where $h$ and $t$ denote the head and tail entities, while $r$ and $\tau$ denote the relation and timestamp, respectively. For each question $q$, we retrieve the top-$k$ relevant KG facts $\mathcal{KG}_k$ from the temporal KGs, which are extracted from the context $c$ via the GPT-4o API[2]:

$$\mathcal{KG}_k = \text{Top-}k(\text{Sim}(q, s_i)), \quad s_i = [h_i; r_i; t_i; \tau_i],$$

where $\text{Sim}(\cdot)$ measures similarity based on either semantic similarity (via Faiss[3] (Douze et al., 2024)) or keyword lexical matching (via KeyBERT(Grootendorst, 2020)). The retrieved KGs are then rephrased into natural language sentences and incorporated into the subsequent LLM input. Full prompt is provided in subsection H.2. The detailed KG extraction pipeline is presented in subsection F.2.

## 4.2 EXPLICIT REASONING INFORMATION EXTRACTION

For explicit guidance, we leverage CoT, which contains correct reasoning steps for each question. Formally, given a question $q$ and its corresponding context $c$, we use the GPT-o1 API to generate paired reasoning outputs $\{r, \hat{a}\}$, where $r$ represents the intermediate reasoning steps and $\hat{a}$ is the final answer. The collected CoT data are used for SFT as a cold start.

---

[1]https://platform.openai.com/docs/models/gpt-4o-mini
[2]https://platform.openai.com/docs/models/gpt-4o
[3]https://github.com/facebookresearch/faiss

Since the cold start does not require a large amount of data, we curate a relatively small but reliable reasoning set (2,119 and 1,045 CoT-annotated samples for TimeQA-Easy and TimeQA-Hard, respectively). To ensure high-quality supervision, during curation we only keep samples where the generated answer $\hat{a}$ matches the ground-truth answer $a$, which constitute a trusted reasoning set. Detailed prompts used for GPT-o1 to generate the CoT data are provided in subsection H.3.

During the CoT-SFT phase, the model is trained to generate both the reasoning steps $r$ and the final answer $a$ obtained from the expert model, effectively learning to perform step-by-step reasoning rather than directly generating answers. After CoT-SFT, the model will be further fine-tuned using RL. The RL training procedure is described in the following subsection 4.3. The whole pipeline of explicit reasoning, information extraction, and subsequent training is presented in Figure 2.

### 4.3 RL TRAINING DETAILS

#### 4.3.1 REWARD MODELING

Rewards are the source of the RL training signal, which controls the optimization direction. Therefore, we design a rule-based reward system consisting of two types of reward:

**Format reward**   We use the format reward to encourage the model to put the thinking process between '`<think>`' and '`</think>`', and final answer between '`<answer>`' and '`</answer>`'. We set the format reward $R_{\texttt{format}} = 0.5$ for pattern matching. Otherwise, it would be $0$.

**Answer reward**   The use of the reward model often suffers from the reward hacking problem (Weng, 2024; Yu et al., 2025). Therefore, following the work on math reasoning problems (Yu et al., 2025; DeepSeek-AI et al., 2025), we design the rule-based outcome reward directly, computed using the rules below:

$$R_{\texttt{ans}}(o, a) = \begin{cases} 1.0, & \text{if } o = \texttt{No Answer} \text{ and } a = \texttt{No Answer} \\ \text{Rouge-L}(o, a) + \text{EM}(o, a), & \text{if } o \neq \texttt{No Answer} \text{ and } a \neq \texttt{No Answer} \\ 0, & \text{Otherwise} \end{cases} \quad (2)$$

where we set the reward to $1.0$ for encouraging the model to be honest if it generates "`No Answer`" when the question cannot be answered. For questions with ground-truths, we choose the Rouge-L score and Exact-Match (EM) as the reward, where $\text{EM}(o, a) = 1.0$ if the generation matches ground-truth, otherwise $0$. Otherwise, we set the reward as $0$ to teach the model to reduce hallucinations (i.e., only $a = \texttt{No Answer}$) and avoid over-abstaining (i.e., only $o = \texttt{No Answer}$). Combining the above two rewards, we get the final reward $R = R_{\texttt{format}} + R_{\texttt{ans}}$ for RL training.

#### 4.3.2 TRAINING TEMPLATE

We use a template in DeepSeek-AI et al. (2025), by designing a straightforward template to guide the model to generate in a specific format. The template is demonstrated in subsection H.6.

## 5 EXPERIMENTS

### 5.1 EXPERIMENTAL SETUP

In this section, we introduce our experimental setup, including the models, training details, datasets, and the evaluation metrics used in the experiments.

**Baselines.**   For baselines, we include both open- and closed-source models in diverse scales: `Qwen2.5-0.5B-Instruct`, `Qwen2.5-1.5B-Instruct`, `Llama3.2-3B-Instruct`, `Qwen2.5-7B-Instruct`, GPT-3.5, and GPT-4o. Recently, Large Reasoning Models (LRMs) have demonstrated substantial reasoning ability improvements, therefore we add two representative LRMs in our experiments: o4-mini and `Qwen3-4B-Thinking`. Further, we train a classifier using `Qwen2.5-1.5B-Instruct`, with a classification head to classify if the question is answerable or not (i.e., 0 for unanswerable and 1 for answerable questions), as a basic baseline. When getting 0 label during evaluation, we set the LLM's answer to "No Answer"; otherwise, we keep the original generated answer from the LLM. We test `Qwen2.5-1.5B-instruct` classifier baseline with `Qwen2.5-7B-Instruct` as the LLM of choice, denoted as Classifier in Table 2.

Table 1: Direct inference (INF) performance of open-source and closed-source LLMs on TimeQA under different implicit reasoning settings, including question-only, original context (C), time-relevant sub-context (Sub-C), and knowledge graphs (chunked or whole). All models are evaluated without any supervised finetuning or reinforcement learning.

| PROMPT TYPE | MODEL TYPE | MODEL | TIMEQA-EASY | | | | | TIMEQA-HARD | | | | |
|---|---|---|---|---|---|---|---|---|---|---|---|---|
| | | | R-1 | R-2 | R-L | BS | EM(%) | R-1 | R-2 | R-L | BS | EM(%) |
| ONLY QUESTION | Open-Source | Qwen2.5-0.5B | 0.062 | 0.052 | 0.062 | 0.309 | 0.00 | 0.074 | 0.062 | 0.074 | 0.313 | 0.00 |
| | | Qwen2.5-1.5B | 0.115 | 0.111 | 0.115 | 0.332 | 11.13 | 0.133 | 0.130 | 0.133 | 0.343 | 13.03 |
| | | Llama3.2-3B | 0.129 | 0.080 | 0.129 | 0.398 | 2.55 | 0.136 | 0.103 | 0.136 | 0.378 | 2.46 |
| | | Qwen2.5-7B | 0.116 | 0.091 | 0.116 | 0.378 | 8.16 | 0.131 | 0.118 | 0.131 | 0.363 | 11.69 |
| | Close-Source | GPT-3.5 | 0.172 | 0.147 | 0.172 | 0.377 | 7.40 | 0.164 | 0.151 | 0.164 | 0.356 | 4.87 |
| | | GPT-4o | 0.310 | 0.253 | 0.310 | 0.490 | 21.56 | 0.263 | 0.227 | 0.263 | 0.450 | 20.72 |
| W/ ORIGINAL CONTEXT (C) | Open-Source | Qwen2.5-1.5B | 0.127 | 0.096 | 0.127 | 0.380 | 8.06 | 0.138 | 0.114 | 0.138 | 0.379 | 10.40 |
| | | Qwen2.5-7B | 0.181 | 0.114 | 0.181 | 0.439 | 8.90 | 0.133 | 0.100 | 0.133 | 0.378 | 9.00 |
| | | Qwen3-4B-Think | 0.328 | 0.271 | 0.271 | 0.371 | 24.56 | 0.278 | 0.221 | 0.277 | 0.333 | 21.51 |
| | Close-Source | GPT-3.5 | 0.443 | 0.364 | 0.442 | 0.608 | 30.18 | 0.303 | 0.256 | 0.303 | 0.494 | 21.09 |
| | | GPT-4o | 0.560 | 0.456 | 0.559 | 0.635 | 39.95 | 0.494 | 0.429 | 0.493 | 0.607 | 29.95 |
| | | o4-mini | **0.634** | **0.519** | **0.632** | **0.711** | **44.14** | **0.531** | **0.439** | **0.530** | **0.639** | **37.12** |
| W/ TIME SUB-C | Close-Source | GPT-3.5 | 0.461 | 0.382 | 0.460 | 0.609 | 31.44 | 0.297 | 0.252 | 0.297 | 0.476 | 18.52 |
| | | GPT-4o | 0.622 | 0.510 | 0.621 | 0.719 | 45.15 | 0.468 | 0.388 | 0.468 | 0.613 | 34.12 |
| W/ KGS (CHUNKED-C) (FAISS) | Close-Source | GPT-3.5 | 0.220 | 0.197 | 0.220 | 0.416 | 15.05 | 0.191 | 0.176 | 0.191 | 0.393 | 13.65 |
| | | GPT-4o | 0.328 | 0.279 | 0.327 | 0.496 | 24.66 | 0.287 | 0.249 | 0.286 | 0.474 | 22.12 |
| W/ KGS (CHUNKED-C) (KEYBERT) | Close-Source | GPT-3.5 | 0.231 | 0.200 | 0.230 | 0.425 | 16.64 | 0.185 | 0.170 | 0.185 | 0.384 | 13.13 |
| | | GPT-4o | 0.338 | 0.288 | 0.338 | 0.505 | 26.72 | 0.233 | 0.202 | 0.233 | 0.392 | 19.10 |
| W/ KGS (WHOLE-C) (FAISS) | Close-Source | GPT-3.5 | 0.283 | 0.247 | 0.283 | 0.467 | 20.12 | 0.211 | 0.194 | 0.211 | 0.407 | 15.46 |
| | | GPT-4o | 0.409 | 0.344 | 0.408 | 0.559 | 30.10 | 0.342 | 0.291 | 0.342 | 0.513 | 26.30 |

Table 2: Performance of models trained with SFT, LoRA, and RL on TimeQA under different reasoning settings. "LoRA" denotes only SFT on LoRA modules (q_proj and v_proj in our study), "Classifier" means training a classification head to classify if the question is answerable.

| PROMPT TYPE | TRAINING METHOD | MODEL | TIMEQA-EASY | | | | | TIMEQA-HARD | | | | |
|---|---|---|---|---|---|---|---|---|---|---|---|---|
| | | | R-1 | R-2 | R-L | BS | EM(%) | R-1 | R-2 | R-L | BS | EM(%) |
| W/ ORIGINAL CONTEXT (C) | SFT | Qwen2.5-1.5B | 0.359 | 0.183 | 0.357 | 0.543 | 1.81 | 0.315 | 0.178 | 0.314 | 0.518 | 3.95 |
| | | Qwen2.5-7B | 0.396 | 0.196 | 0.396 | 0.564 | 3.70 | 0.370 | 0.180 | 0.365 | 0.540 | 0.37 |
| | LoRA (r=32) | Qwen2.5-1.5B | 0.321 | 0.184 | 0.316 | 0.519 | 1.29 | 0.246 | 0.153 | 0.243 | 0.467 | 2.37 |
| | Classifier | Qwen2.5-7B | 0.159 | 0.126 | 0.158 | 0.387 | 11.41 | 0.138 | 0.128 | 0.137 | 0.357 | 12.40 |
| | RL | Qwen2.5-0.5B | 0.469 | 0.355 | 0.468 | 0.660 | 32.48 | 0.411 | 0.310 | 0.410 | 0.626 | 25.26 |
| | | Qwen2.5-1.5B | **0.580** | **0.456** | **0.578** | **0.722** | **43.41** | **0.504** | **0.395** | **0.504** | **0.681** | **35.75** |
| W/ TIME SUB-C | SFT | Qwen2.5-1.5B | 0.442 | 0.283 | 0.440 | 0.587 | 1.08 | 0.311 | 0.184 | 0.311 | 0.524 | 1.41 |
| | | Qwen2.5-7B | 0.484 | 0.314 | 0.483 | 0.552 | 1.01 | 0.374 | 0.232 | 0.373 | 0.552 | 1.96 |
| W/ KGS (CHUNKED C) (FAISS) | SFT | Qwen2.5-1.5B | 0.238 | 0.151 | 0.236 | 0.463 | 2.50 | 0.225 | 0.120 | 0.225 | 0.470 | 0.13 |
| | | Qwen2.5-7B | 0.271 | 0.154 | 0.270 | 0.496 | 0.43 | 0.271 | 0.164 | 0.269 | 0.484 | 0.32 |
| W/ KGS (WHOLE C) (FAISS) | SFT | Qwen2.5-1.5B | 0.285 | 0.159 | 0.284 | 0.490 | 0.07 | 0.267 | 0.180 | 0.267 | 0.462 | 0.39 |
| | | Qwen2.5-7B | 0.322 | 0.198 | 0.320 | 0.511 | 1.40 | 0.310 | 0.174 | 0.307 | 0.507 | 0.26 |

**Training Settings.** We include four LLMs in different sizes: `Qwen2.5-0.5B-Instruct`, `Qwen2.5-1.5B-Instruct` (Qwen et al., 2025), `Llama3.2-3B-Instruct` (Meta, 2024) and `Qwen2.5-7B-Instruct` (Qwen et al., 2025), which have been instruction-tuned to improve their reasoning and instruction-following abilities. We conduct all training experiments on 4xH100 NVIDIA GPUs. In Supervised Fine-tuning (SFT) experiments, we conduct full-parameter fine-tuning on the original full data. The model is fine-tuned in 2 epochs, with learning rate $lr = 1e^{-5}$, weight decay $wd = 1e^{-2}$. In Reinforcement Learning (RL), we first SFT the model on CoT data for 1 epoch, keeping the same as the SFT experiments. Then we train via GRPO for 3 epochs, with learning rate $lr = 1e^{-5}$, group size $G = 4$, KL coefficient $\beta = 0.01$, and $\epsilon = 0.2$.

**Temporal QA Datasets.** To evaluate the temporal reasoning with the abstention capability of LLMs, we use TimeQA (Chen et al., 2021), a benchmark specifically designed for time-sensitive question answering. Unlike traditional QA datasets, TimeQA requires reasoning over temporal scopes. TimeQA contains over 20,000 question-answer pairs, covering 5.5K time-evolving facts and 70 distinct relations. Each example includes a long document context and a time-sensitive query, where the correct answer depends on the query time and the temporal scope of relevant facts. TimeQA consists of two datasets with two difficulty levels of easy and hard to evaluate temporal understanding and reasoning as shown in Appendix G: (1) Easy questions tend to align with explicit time expressions (e.g., one specific time). (2) Hard questions involve implicit or abstract temporal references (e.g., "early 1980s") requiring deeper reasoning. The datasets include both answerable and unanswerable examples, allowing assessment of models' ability to abstain when no valid answer exists.

**Non-Temporal (Out-Of-Distribution, or OOD) QA Datasets.** To evaluate the generality of LLMs' abstention behavior beyond temporal questions, we additionally include three non-temporal Multiple-Choice QA datasets in our experiments: MMLU (Hendrycks et al., 2020), HellaSwag (Zellers et al., 2019), and SQuAD v2 (Rajpurkar et al., 2018). These datasets cover a wide range of domains and reasoning types but do not require explicit temporal understanding. Details of dataset statistics and pre-processing steps are provided in Appendix L.

**Evaluation Metrics.** As our task is text generation, we measure performance on both lexical overlap metrics and semantic matching. For the lexical level, we select ROUGE (Lin, 2004) for automatic evaluation between model outputs and ground-truth answers, to measure the quality of the model's generated texts. For the semantic level, we leverage BERTScore (denoted as BS.) (Zhang et al., 2020), which assesses the semantic similarity between a candidate text (e.g., a machine translation or summary) and a reference text. We include Exact-Match (EM) as a metric as well, to observe how precisely the model generates.

Additionally, to quantify the abstention performance on unanswerable questions, we report the number of samples classified as True Positive (TP), False Positive (FP), and False Negative (FN) as metrics, which are defined as follows for this specific task: TP: $o = $ `No Answer`, $a = $ `No Answer`; FP: $o = $ `No Answer`, $a \neq $ `No Answer`; FN: $o \neq $ `No Answer`, $a = $ `No Answer`.

## 5.2 MAIN RESULTS

In this section, we systematically evaluate models of different scales, input information types, and training paradigms (SFT vs. RL). The results are shown in Table 1 and Table 2. These experiments would help identify the most effective configurations for the temporal question answering task.

**1. LLM variants:** In the direct inference (INF) setting without any contexts provided (i.e., only given questions), the performances of both open- and closed-source models are not satisfactory. For the closed-source models, we use the prompts described in subsection H.4 and subsection H.5. The performances improve roughly as the size of the model increases. It is noteworthy that even for one of the most state-of-the-art models like GPT-4o on TimeQA-Hard, it only achieves $20.72\%$ on Exact-Match, which indicates the difficulty of this task and the weakness of the current LLMs.

**2. Implicit information variants:** We examine the effects of different types of implicit information, including original contexts $c$, time-relevant sub-context $tc$, and KGs. For extracting KGs from contexts, we try two methods as well. One is feeding the whole original contexts into GPT-4o-mini directly (denoted as WHOLE-CONTEXT); another way is segmenting the original contexts first into several sub-contexts, then extracting KGs from each sub-context (denoted as CHUNKED-CONTEXT).

**W/ Original Context $c$:** With the original context $c$ added into the prompt, we observe significant improvements compared to only asking question $q$ among models, on the TimeQA-Easy dataset. For example, `Qwen2.5-7B-Instruct` achieves $0.181$ from $0.116$ in R-1 score. However, on the TimeQA-Hard the improvements are less noticeable, which also implies that Hard-version questions are too challenging to reason even with the original contexts. Further, after supervised fine-tuning, there is a significant enhancement in the ROUGE score, while EM drops, which may imply that SFT encourages the model to generate answers similar to the ground-truths, not exact matching. The results of the binary classifier baseline show that it does not perform well for predicting whether a temporal question is answerable or not. Moreover, we observe that reasoning models outperform non-reasoning models (e.g., o4-mini vs. GPT-4o) substantially.

**W/ Time-relevant Sub-Context $tc$:** It's surprising to observe that in TimeQA-Easy, with the time-relevant sub-context $tc$ the model performs better than even using the original context, which indicates that extracted relevant information can boost the model's performance better. The pattern is also similar in Hard set (e.g., from $39.95$ to $45.15$ in EM for GPT-4o). From this experiment, we conclude that compared to the original context, it is more recommended to use $tc$ without unnecessary additional information, which leads to a large margin improvement.

**W/ KGs:** We compare two different approaches to extract KGs: one is from WHOLE-CONTEXT, and the other is from CHUNKED-CONTEXT. Although chunked-context yields more fine-grained KGs, it may lose continuity, which explains why the former method outperforms the latter. In the chunked setting, we experiment with two similarity-based extraction methods (Faiss-based and KeyBERT-based methods), and observe that both yield comparable performance, while the Faiss-based method can perform better than KeyBERT on TimeQA-Hard. However, KGs perform less

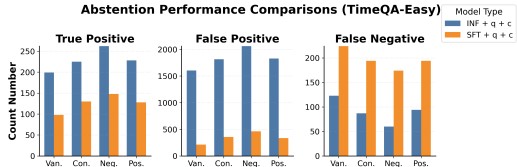 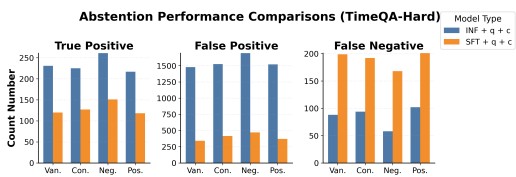

(a) Abstention performances on TimeQA-Easy.   (b) Abstention performances on TimeQA-Hard.

Figure 3: Abstention performance comparisons across various prompts on TimeQA task with `Qwen2.5-1.5B-Instruct` and original context $c$. Van., Con., Pos., and Neg. denote for VANILLA, CONTRASTIVE, POSITIVE, NEGATIVE Prompts, respectively.

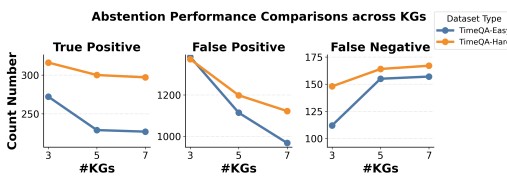 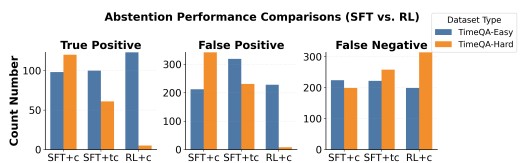

(a) Abstention performance comparisons across different numbers of KGs on TimeQA with `Qwen2.5-1.5B-Instruct`.

(b) Abstention performance comparisons between SFT and RL on TimeQA with `Qwen2.5-1.5B-Instruct`.

Figure 4: Abstention performance comparisons across different numbers of KGs (a) and different training methods (b).

effectively than using the contexts $c$ or $tc$, suggesting that simply putting the context in the prompt would be the most efficacious method for practical application.

**3. RL vs. SFT:** In addition to standard SFT, we further explore if RL can improve the model's performance, based on the previous success on math (DeepSeek-AI et al., 2025).

**RL with a CoT-SFT cold start can unlock the best reasoning capability even for the smallest model, with a simple reward design.** For RL training, we conduct full-parameter tuning on `Qwen2.5-1.5B-Instruct`. The most surprising observation is that with RL, the 1.5B model can achieve the best results, even outperforming GPT-4o. For instance, its EM score outperforms GPT-4o by $3.46\%$ and $5.80\%$, on TimeQA-Easy and Hard, respectively. However, it should be noted that when using the base model (i.e. without CoT-SFT), RL training fails, implying that CoT-SFT is essential for incentivizing inherent capabilities, with step-by-step guidance from an expert model.

**Implicit clues at the SFT stage cannot improve reasoning ability.** Even full-parameter tuning `Qwen2.5-7B-Instruct` with $c$, $tc$ or KGs, the performances are much worse than `Qwen2.5-1.5B-Instruct` under RL training. These results strongly demonstrate that SFT alone cannot teach the model how to reason in the temporal reasoning with abstention task, and the improvement from SFT is limited, even when tuning a larger model.

## 6 ANALYSIS

### 6.1 EFFECTS OF VARIOUS PROMPTS DESIGN

Prompts play an essential role in effectively structuring, evaluating, and performing diverse tasks (Schulhoff et al., 2025). We evaluate the model's abstention performances under four different prompts: VANILLA PROMPT, CONTRASTIVE PROMPT, POSITIVE PROMPT, and NEGATIVE PROMPT, with details provided in subsection H.7. We compare these prompts' effects on `Qwen2.5-1.5B-Instruct` under the setting using the original context $c$ shown in Figure 3.

**NEGATIVE PROMPT would cause the model to be "over-prudent".** Compared to the other three types of prompts, NEGATIVE PROMPT mostly increases the TP and FP, while decreasing FN. This implies that the "negative examples" can push the model to abstain more as a hint. Another interesting finding is that POSITIVE PROMPT performs almost the same as CONTRASTIVE PROMPT. Even when providing "positive examples" in the prompt, it can still promote the model to generate "`No Answer`" on the TimeQA-Easy dataset.

## 6.2 Effects of the number of KGs

We first investigate the effect of the number of KGs in the prompts, to explore how the model would behave when providing more KGs. The result is presented in Figure 4a.

**With more KGs, the model abstains less frequently.** On both TimeQA datasets, as the number of KGs increases, the model becomes more "confident" and generates "`No Answer`" less (e.g., a dramatic drop in the False Positive). This phenomenon poses a practical dilemma: More relevant KGs provided may enhance the performance on question answering (referring to Table 12 in Appendix I), but there exists a risk of degrading the abstention capability.

## 6.3 Effects of training methods

**SFT makes the model "overconfident".** Based on Figure 3, SFT shifts models from over-cautious to over-confident behavior. Compared to direct inference (INF), SFT reduces FP but also lowers TP and raises FN. Even when trained for abstention, fine-tuned models still overproduce answers to unanswerable questions, suggesting that SFT enhances coverage more than genuine uncertainty recognition and thus compromises reliability.

**LLMs display contrasting abstention patterns after RL.** In Figure 4b, on TimeQA-Easy, RL boosts honesty more significantly than SFT, by increasing true positives and lowering hallucinations. However, on TimeQA-Hard it reverses: the model avoids abstention and sharply increases false negatives, probably because abstaining correctly in complex temporal reasoning scenarios is harder than producing speculative answers.

## 6.4 Effects of Hyperparameters

We further explore the effects of hyperparameters including $\beta$ in Equation 1, diverse reward designs (in Table 3) and data distribution in Table 14 of Appendix K.

**Lower $\beta$ might be better.** As Yu et al. (2025) pointed out, long-CoT training risks distributional drift, motivating the removal of the KL penalty ($\beta = 0.0$). Our results confirm this sensitivity.

Table 3: Effects of $\beta$ and different rewards of RL training on TimeQA-Easy, with `Qwen2.5-1.5B-Instruct`.

| | TimeQA-Easy | | | TimeQA-Hard | | |
|---|---|---|---|---|---|---|
| | R-1 | BS | EM(%) | R-1 | BS | EM(%) |
| RL ($\beta = 0.01$, $R_{ans}$=ROUGE+EM) | 0.580 | 0.722 | 43.41 | 0.504 | 0.681 | 35.75 |
| RL ($\beta = 0.01$, $R_{ans}$=ROUGE) | 0.589 | 0.725 | 42.74 | 0.446 | 0.641 | 26.67 |
| RL ($\beta = 0.04$, $R_{ans}$=ROUGE) | 0.576 | 0.720 | 41.56 | 0.432 | 0.640 | 25.40 |
| RL ($\beta = 0.01$, $R_{ans}$=ROUGE+BERTScore) | 0.505 | 0.679 | 34.40 | 0.508 | 0.683 | 35.46 |
| RL ($\beta = 0.01$, $R_{ans}$=ROUGE+BERTScore+EM) | 0.472 | 0.665 | 33.71 | 0.523 | 0.689 | 37.62 |

**Reward design is delicate**. Using ROUGE alone hurts performance, especially on Hard cases where lexical overlap fails to capture temporal reasoning. Replacing EM with BERTScore lowers accuracy on Easy, as semantic similarity may encourage plausible but incorrect answers to hack the rewards. Combining all three rewards helps on Hard by complementing sparse EM signals, but sharply degrades Easy performance, where questions are easier and EM alone is already reliable.

**Data ratio matters.** On the original imbalanced TimeQA-Easy set (only $12.4\%$ unanswerable), the model largely ignores abstention. However, after increasing the proportion of unanswerable questions in the dataset to 50%, the model collapses into "always abstain". This demonstrates that robust abstention requires both controlling the data distribution and carefully adjusting the reward design (more details can be found in Appendix K).

## 6.5 Generalization Analysis

**Generalize from temporal QA to OOD QA.** We first investigate whether models trained on TimeQA can generalize their abstention capability to non-temporal (OOD) tasks. In Table 4, after SFT on TimeQA-Easy, both 1.5B and 7B models show consistent improvements in true positives, suggesting a partial acquisition of abstention ability. This effect is weaker after SFT on TimeQA-Hard, with improvements observed only for the 7B model. However, RL training induces overconfidence, consistent with the trends observed in Figure 4b. All experiments illustrate that abstention ability is difficult to generalize to out-of-distribution domains (case studies are in Appendix M).

**Generalize between TimeQA-Easy and Hard.** We then explore generalization between two subsets of TimeQA, to investigate if the reasoning ability can also be generalized between different diffi-

Table 4: Generalization experiments from TimeQA to OOD datasets, for evaluating the model's abstention ability.

| | HellaSwag | | | MMLU | | | SQuAD | | |
|---|---|---|---|---|---|---|---|---|---|
| | TP | FP | FN | TP | FP | FN | TP | FP | FN |
| Qwen2.5-1.5B (INF) | 36 | 267 | 204 | 18 | 103 | 222 | 1266 | 1047 | 265 |
| Qwen2.5-7B (INF) | 7 | 44 | 233 | 17 | 60 | 223 | 193 | 19 | 1338 |
| Qwen2.5-1.5B (TimeQA-Easy, SFT+$c$) | 43 | 161 | 197 | 28 | 133 | 212 | 1052 | 665 | 479 |
| Qwen2.5-7B (TimeQA-Easy, SFT+$c$) | 33 | 134 | 207 | 42 | 92 | 199 | 1062 | 330 | 469 |
| Qwen2.5-1.5B (TimeQA-Hard, SFT+$c$) | 20 | 136 | 220 | 16 | 119 | 224 | 889 | 578 | 642 |
| Qwen2.5-7B (TimeQA-Hard, SFT+$c$) | 8 | 51 | 232 | 18 | 31 | 222 | 1031 | 359 | 500 |
| Qwen2.5-1.5B (TimeQA-Easy, RL+$c$) | 5 | 1 | 235 | 8 | 1 | 232 | 11 | 4 | 1520 |
| Qwen2.5-1.5B (TimeQA-Hard, RL+$c$) | 2 | 9 | 238 | 1 | 7 | 239 | 1 | 1 | 1530 |

Table 5: Generalization between TimeQA-Easy and Hard, with `Qwen2.5-1.5B-Instruct`.

| | TimeQA-Easy | | TimeQA-Hard | |
|---|---|---|---|---|
| | BS | EM(%) | BS | EM(%) |
| IN-DISTRIBUTION (SFT+$c$) | 0.543 | 1.81 | 0.518 | 3.95 |
| OUT-DISTRIBUTION (SFT+$c$) | 0.580 | 3.98 | 0.514 | 2.66 |
| IN-DISTRIBUTION (RL+$c$) | 0.725 | 42.74 | 0.641 | 26.67 |
| OUT-DISTRIBUTION (RL+$c$) | 0.643 | 27.88 | 0.642 | 29.84 |

culties. The results are shown in Table 5. IN-DISTRIBUTION means training and testing on the same dataset, while OUT-DISTRIBUTION denotes cross-dataset evaluation (e.g., OUT-DISTRIBUTION on TimeQA-Easy means training on Hard and testing on Easy).

Surprisingly, under RL, training on Hard does not improve performance on Easy, whereas training on Easy performs effectively on Hard. This likely reflects the 1.5B model's limited ability to extract useful reward signals from the more challenging Hard set. In contrast, on SFT, generalizing from Hard to Easy can outperform in-distribution training (e.g., $+2.17\%$ EM), indicating that supervised signals from Hard are more robust for cross-difficulty generalization.

## 7  CONCLUSIONS AND KEY TAKEAWAYS

We present the first systematic study of abstention-aware temporal QA with LLMs, where models must reason over time-sensitive contexts and recognize when no valid answer exists. We studied a variety of training methods as well as input information types through extensive experiments. Our findings demonstrate that reasoning with abstention ability on temporal QA remains challenging for LLMs, even for the state-of-the-art models. In particular, our model initialized with Qwen2.5-1.5B-Instruct can outperform GPT-4o on reasoning with abstention using our training pipeline, demonstrating that RL, especially when paired with explicit CoT supervision, contributes more to robust generalization than additional supervised fine-tuning alone. Yet, despite acquiring in-distribution reasoning and abstention skills, models generalize poorly across domains. This underscores the need for more principled uncertainty-aware training and evaluation strategies to build LLMs that are not only better reasoners but also more reliable in knowing when to abstain.

---

**KEY TAKEAWAYS**

1. Abstention in temporal QA tasks is a learnable skill which could be guided with RL.

2. Starting RL from SFT with CoTs equips smaller models with core reasoning with abstention skills, enabling them to rival or surpass larger closed-source models. (subsection 5.2)

3. Contexts and KGs provide limited improvement compared with CoT. (subsection 5.2)

4. SFT can induce overconfident behavior in LLMs; RL can enhance reasoning with abstention ability, but the risk of overconfidence still exists. (subsection 6.3)

5. Determining the optimal proportion of unanswerable data in the training set is a key challenge for robust abstention. (Appendix K)

6. Abstention ability is difficult to generalize to out-of-distribution domains. (subsection 6.5)

---

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

# Appendix

## Table of Contents

## A    LIMITATIONS

While our study advances abstention-aware temporal QA, several limitations remain. First, our experiments focus on medium-scale open-source models ($\leq$7B parameters) due to computational resource constraints and a limited set of benchmarks; this encourages the development of new large-scale or domain-specific (e.g., biomedical, legal) datasets for the abstention task. Second, our work only focuses on the English language, which would be interesting to see if further generalization to multilingual settings is viable. Finally, our work primarily evaluates single-turn QA. In multi-turn dialogue or interactive reasoning, abstention may require additional dynamics, such as explaining refusals or maintaining consistency across turns.

## B    REPRODUCIBILITY STATEMENT

We include all the prompts we use in the Appendix H for the convenience of future research and transparency of the community. Our experiments are based on HuggingFace TRL[4] library, which is open-source and designed for post-training foundation models.

## C    THE USE OF LARGE LANGUAGE MODELS (LLMS)

We only leverage some use of LLMs for polishing the content of our paper.

## D    EXTENDED RELATED WORK

**SFT and RL in LLMs.**    Post-training is essential for enhancing LLMs' performance (Zhang et al., 2022; OpenAI et al., 2024; Meta, 2024), commonly applying large-scale SFT (Radford et al., 2018; 2021; Chung et al., 2022; Zhou et al., 2023), and RL (Ouyang et al., 2022a; Sun et al., 2023; Zhou et al., 2024; Zhai et al., 2024). SFT often utilizes instruction-formatted data to adapt the pre-trained LLMs to specific downstream tasks. For example, LIMA (Zhou et al., 2023) demonstrates that SFT can adapt the model's generated responses to a desired format effectively (Li et al., 2025). Some examples of this even go back to the pre-LLM era on controlled text generation (Bahrainian et al., 2021) by using topics (i.e. high-level concepts) (Ravenda et al., 2025; Bahrainian & Crestani, 2016) as a guidance signal under SFT to control (temporal) topics (Bahrainian et al., 2018) or sentiment (Bahrainian & Dengel, 2015; Bahrainian et al., 2014) of generated texts. By contrast, RL (Ouyang et al., 2022a; Sun et al., 2023; Zhai et al., 2024; Zhou et al., 2024) has been used to align models with human preferences to ensure safety. Recently, RLVR has attracted researchers as a method to improve the reasoning ability of LLMs in domains such as mathematics and coding (Shao et al., 2024; Lambert et al., 2025; Yang et al., 2025; Liang et al., 2025). However, there still remains a lack of a deeper understanding of the function and limitations of RL. Diverse studies (Liu et al., 2025; Zhao et al., 2025; Yue et al., 2025) show that the reflective behaviors in DeepSeek-R1-like models actually emerge from the base model, instead of being learned through RLVR training. However, these works mainly focus on the mathematical reasoning tasks. Our paper shifts the attention to the general reasoning task involving both non-temporal/temporal reasoning with abstention, which has not been systematically studied previously.

**Temporal Question Answering Task.**    Temporal QA poses unique challenges beyond standard QA, requiring models to reason over dynamic events, temporal relationships, and evolving world states. Recent benchmarks have aimed to evaluate and enhance temporal reasoning in LLMs. Tan et al. (2023) introduce a comprehensive benchmark for multi-level temporal reasoning. Chen et al. (2021) propose Time-Sensitive QA, where models must handle time-evolving facts and abstain when temporal context is lacking. Mavromatis et al. (2022) develop TempoQR, which augments question representations with time-aware and entity-based signals to improve reasoning over temporal knowledge graphs. Ding et al. (2023b) focus on future-oriented QA, using past Knowledge Graph (KG) snapshots to predict future events. Wei et al. (2025) design a comprehensive benchmark TIME that captures the complexity of temporal reasoning in diverse real-world scenarios, including knowledge-intensive, dynamic events, and multi-session interactive contexts. Our work complements these

---

[4]https://github.com/huggingface/trl

efforts by focusing on a critical yet underexplored aspect: how well LLMs can abstain from answering when temporal information is ambiguous or missing, and whether LLMs can transfer the ability acquired from temporal problems to the non-temporal domain.

## E    DETAILS OF GRPO

We include the detailed definition of Equation 1 as follows:

$$A_i = \frac{r_i - \texttt{mean}(\{r_1, r_2, ..., r_G\})}{\texttt{std}(\{r_1, r_2, ..., r_G\})}$$

$$\mathbb{D}_{\text{KL}}(\pi_\theta || \pi_{ref}) = \frac{\pi_{ref}(\boldsymbol{o}^{(i)}|\boldsymbol{q})}{\pi_\theta(\boldsymbol{o}^{(i)}|\boldsymbol{q})} - \log \frac{\pi_{ref}(\boldsymbol{o}^{(i)}|\boldsymbol{q})}{\pi_\theta(\boldsymbol{o}^{(i)}|\boldsymbol{q})} - 1$$

## F    DETAILS OF IMPLICIT REASOINING INFORMATION EXTRACTION

We categorize background knowledge and context signals that indirectly support reasoning as implicit reasoning information. In this work, we focus on two main types of implicit reasoning information: time-related information and Knowledge Graphs (KGs). The extracted implicit reasoning information will be concatenated with the original question and provided as input to the LLM.

### F.1    TIME-RELATED INFORMATION EXTRACTION

In temporal question answering, temporal cues in the supporting context often play a critical role in locating the correct answer. On the other hand, incorrect and irrelevant temporal information would confuse the model. Therefore, we design a Time-related Information Extraction module that automatically identifies and filters time-relevant sub-contexts $\boldsymbol{tc}$ to better support question answering.

Given a natural language question $q_i$ associated with a timestamp or time interval $t_{q_i}$, and a background context $c_i$ containing potentially relevant temporal knowledge and other irrelevant information, we employ GPT-4o-mini[5], which is a fast model for focused tasks, to extract sub-contexts $tc_i \subset c_i$ that are temporally aligned with $t_{q_i}$ with a prompt instructing the model to retain only sentences or facts that relate to the time scope of the question.

This temporal sub-context is then concatenated with the original question and passed to the LLM during inference. In this way, we ensure that the model receives temporally aligned evidence without being distracted by unrelated background facts.

### F.2    KG EXTRACTION

In addition to the above straightforward extraction, we design a KG Extraction module that retrieves relevant facts from a temporal knowledge graph (TKG) for each input question. As illustrated in Figure 5, we implement two distinct extraction strategies: a semantic similarity-based retrieval using Faiss [6] (Douze et al., 2024) and a lexical matching-based retrieval using KeyBERT (Grootendorst, 2020).

**Semantic Similarity Retrieval via Faiss**   In the first strategy, we concatenate the head entity, relation, tail entity, and timestamp of each KG quadruple into a single KG sentence. These KG sentences, along with the question, are then encoded into a shared vector space using a sentence encoder (e.g., Sentence-BERT (Reimers & Gurevych, 2019)). After being converted into dense embeddings, **Faiss** is used to efficiently retrieve Top-$k$ similar KG facts ($k = 10$ in our experiments). This semantic path excels at capturing paraphrased relations or latent temporal associations, even when there is no exact lexical overlap between the question and the KG entries.

**Keyword Lexical Matching via KeyBERT**   The second strategy focuses on explicit lexical matching. We use **KeyBERT** (Grootendorst, 2020) to extract salient keywords from the input question, and match these against components of each KG quadruple: namely the head entity, relation, tail entity, and timestamp. Candidate quadruples are scored based on the number of keyword overlaps. The top-matching quadruples are selected as the most lexically relevant knowledge context.

---

[5]https://platform.openai.com/docs/models/gpt-4o-mini
[6]https://github.com/facebookresearch/faiss

This method is effective for precision-focused retrieval when the question uses terminology that directly aligns with the KG structure.

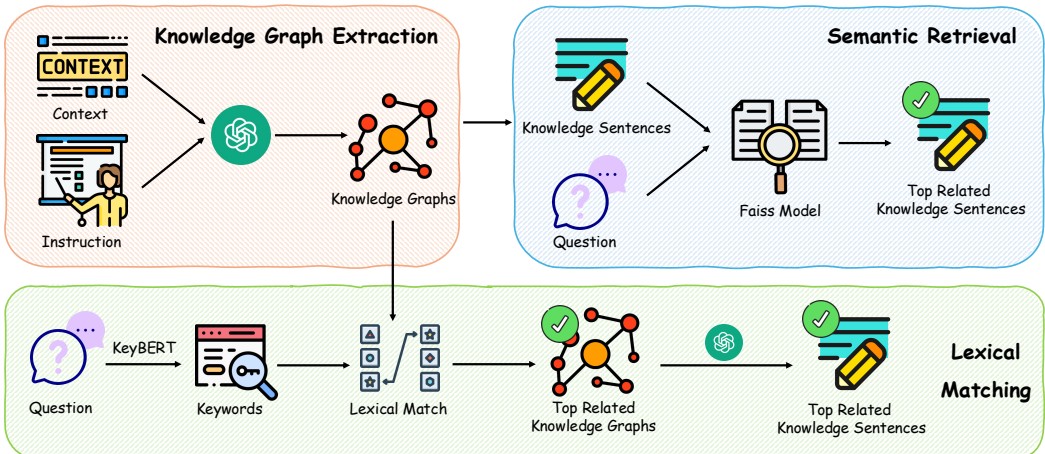

Figure 5: The KG-related information extraction pipeline. Knowledge Graphs are extracted by GPT-4o-mini given the contexts, which are ranked in either a semantic-based or a lexical-based way.

## G TimeQA Descriptions

We provide the examples from both Easy and Hard set of TimeQA, shown in Table 6, for a basic understanding.

Table 6: Examples from the easy version and hard version of TimeQA datasets showing temporal reasoning requirements and answerability.

| Question | Answer | Time Specifier | Difficulty | Answerable |
|---|---|---|---|---|
| Which title was conferred to Jorge Cori in 2009? | International Master | in 2009 | Easy | Yes |
| Who was the spouse of Anna Karina from 1968 to 1974? | Pierre Fabre | 1968–1974 | Easy | Yes |
| Carl Eric Almgren took which position after Oct 1969? | Chief of the Army | after Oct 1969 | Hard | Yes |
| What was the family name of Paula Hitler after Apr 1957? | [unanswerable] | after Apr 1957 | Hard | No |

## H Various Prompts Design

### H.1 Time-related Sub-context Extraction Prompt

The prompt for time-related sub-context extraction is shown in Table 7.

### H.2 KG Extraction Prompt

The prompt for time-related sub-context extraction is shown in Table 8.

### H.3 Chain-of-Thoughts Generation Prompt

The prompt for generating chain-of-thought reasoning texts using GPT-o1 API is shown in Figure 6.

### H.4 Closed-source Models' Prompt

The prompt for GPT-3.5, GPT-4o is shown in Table 9.

Table 7: Template of for time-related sub-context extraction.

---

**Time-related Sub-context Extraction Prompt**

---

You are a top-tier algorithm designed for extracting sentences with time information in the text. Try to capture as much time information from the text as possible without sacrificing accuracy. Do not add any information that is not explicitly mentioned in the text. Your task is to identify the complete sentences with time information requested with the user prompt from a given text related to the given query. You must generate the output in a JSON format containing a list of complete sentences with time information from the given text.

IMPORTANT NOTES:
- Don't add any explanation and text. Don't change the original sentence.
- Identify all the events or actions that have time-related details.

Query: `<question>`
Context: `<context>`
assistant:

---

Table 8: Template of for KG extraction.

---

**KG Extraction Prompt**

---

You are a top-tier algorithm designed for extracting information in structured formats to build a knowledge graph. Try to capture as much information from the text as possible without sacrificing accuracy. Do not add any information that is not explicitly mentioned in the text. Your task is to identify the entities and relations and timestamps requested with the user prompt from a given text. You must generate the output in a JSON format containing a list with JSON objects. Each object should have the keys: "head", "head_type", "relation", "tail", "tail_type" and "timestamp". The "head" key must contain the text of the extracted entity. The "head_type" key must contain the type of the extracted head entity, The "relation" key must contain the type of relation between the "head" and the "tail". The "tail" key must represent the text of an extracted entity which is the tail of the relation, and the "tail_type" key must contain the type of the tail entity. The "timestamp" key must contain the timestamp of the event if it is present in the text. If the timestamp is not present, the value of the "timestamp" key must be null. Your task is to extract relationships from text strictly adhering to the provided schema. The relationships can only appear between specific node types are presented in the schema format like: (Entity1Type, RELATIONSHIP_TYPE, Entity2Type, TIME) /n Attempt to extract as many entities and relations as you can. Maintain Entity Consistency: When extracting entities, it's vital to ensure consistency. If an entity, such as "John Doe", is mentioned multiple times in the text but is referred to by different names or pronouns (e.g., "Joe", "he"), always use the most complete identifier for that entity. The knowledge graph should be coherent and easily understandable, so maintaining consistency in entity references is crucial. Identify all the events or actions that have time-related details.
IMPORTANT NOTES:
- Don't add any explanation and text.

Query: `<question>`
Context: `<context>`
assistant:

---

## H.5 LARGE REASONING MODEL (LRM)'S PROMPT

The prompt for o4-mini and Qwen3-4B-Thinking is shown in Table 10.

## H.6 RL TRAINING PROMPT

The prompt for RL training is shown in Table 11.

You are an expert in generating high-quality thinking and reasoning data.
Your task is to create detailed, step-by-step reasoning processes that demonstrate clear logical thinking and problem-solving abilities.
Follow these guidelines:

1. Structure and Format:
- Start with a clear problem statement or question
- Break down the problem into smaller, manageable components
- Present your reasoning in a step-by-step manner
- Use clear and precise language
- Include relevant assumptions and their justifications
- Conclude with a well-reasoned answer

2. Quality Requirements:
- Demonstrate logical consistency throughout the reasoning process
- Consider multiple perspectives when relevant
- Acknowledge potential limitations or edge cases
- Use concrete examples to illustrate abstract concepts
- Show how you arrive at conclusions from premises
- Include intermediate steps and thought processes

3. Reasoning Components:
- Identify key concepts and their relationships
- Apply relevant principles or rules
- Show how you evaluate different options
- Explain why certain approaches are chosen over others
- Demonstrate critical thinking and analysis
- Consider implications and consequences

4. Output Format:
<think>
Problem: [Clear statement of the problem/question]

Analysis:
1. [First step of reasoning]
    - Sub-point or explanation
    - Supporting evidence or logic

2. [Second step of reasoning]
    - Sub-point or explanation
    - Supporting evidence or logic

[Continue with additional steps as needed]

Considerations:
- [Important factor 1]
- [Important factor 2]
- [Potential limitations or assumptions]
</think>

<answer>
[direct answer entity without any explanation]
</answer>

5. Additional Guidelines:
- Maintain objectivity in your reasoning
- Use precise and unambiguous language
- Show your work and explain your thought process
- Consider both short-term and long-term implications
- Acknowledge uncertainty when present
- Provide context when necessary

Remember to:
- Be thorough but concise
- Show your work clearly
- Justify your reasoning
- Consider multiple perspectives
- Draw clear conclusions
- Maintain logical consistency

You will be given temporal questions and context related the question. Reason using the context.
Your responses should demonstrate expert-level thinking and reasoning while remaining accessible and understandable to the reader.

Figure 6: The prompt for generating chain-of-thought reasoning texts using GPT-o1 API.

Table 9: Template of GPT models for evaluation.

| **GPT's Prompt** |
| --- |
| You are a top expert at answering questions about time. You have to answer the questions without any explanations according to the context. If there is no answer towards the question in the context, please respond "no answer". Do not search the website. |

Table 10: Template of large reasoning models for evaluation.

| **LRM's Prompt** |
| --- |
| You are a top expert at answering questions about time. Please think step by step, and give the answer between `<answer>` and `</answer>` according to the information provided. If there is no answer towards the question in the context, please respond "No Answer". Do not search the internet and website. |

Table 11: Template of Qwen for RL training. prompt will be replaced with the specific question (with different types information in different settings) during training.

| **RL Training Prompt** |
| --- |
| system: You are Qwen, created by Alibaba Cloud. You are a helpful assistant. A conversation between User and Assistant. The user asks a question, and the Assistant solves it based on the given information. The assistant first thinks about the reasoning process in the mind and then provides the user with the answer within 80 words. If there is no correct answer to the question, print No Answer. The reasoning process and answer are enclosed within `<think> </think>` and `<answer> </answer>` tags, respectively, i.e., `<think>` reasoning process here `</think><answer>` answer here `</answer>`. user: prompt. assistant: |

## H.7 DIVERSE PROMPTS DESIGN FOR IN-CONTEXT LEARNING

We provide the details of four different prompts in `Qwen-2.5` models as follows:

**VANILLA PROMPT:** `"You are Qwen, created by Alibaba Cloud. You are a helpful assistant.`
`Think and give the correct answer of the following question without any other explanation based on the given context.`
`If there is no correct answer to the question, print No Answer."`

**CONTRASTIVE PROMPT:** `"You are Qwen, created by Alibaba Cloud. You are a helpful assistant.`
`Think and give the correct answer of the following question without any other explanation based on the given examples and context.`
`Examples:`
`Question: Which team did George Moorhouse play for from 1921 to 1923? Answer: Tranmere Rovers`
`Question: What was the place of detention for Josep Rull from Jun 2019 to Jun 2020? Answer: No Answer`
`If there is no correct answer to the question, print No Answer."`

**POSITIVE PROMPT:** `"You are Qwen, created by Alibaba Cloud. You are a helpful assistant.`
`Think and give the correct answer of the following question without any other explanation based on the given examples and`

```
context.
Examples:
Question:  Which team did George Moorhouse play for from 1921 to
1923?  Answer:  Tranmere Rovers
If there is no correct answer to the question, print No Answer."
```

**NEGATIVE PROMPT:** `"You are Qwen, created by Alibaba Cloud.  You are`
`a helpful assistant.`
`Think and give the correct answer of the following question`
`without any other explanation based on the given examples and`
`context.`
`Examples:`
`Question:  What was the place of detention for Josep Rull from Jun`
`2019 to Jun 2020?  Answer:  No Answer`
`If there is no correct answer to the question, print No Answer."`

For POSITIVE PROMPT, we include a QA pair which has a ground-truth; and for NEGATIVE PROMPT, we include a QA pair which cannot be answered; for CONTRASTIVE PROMPT, we include both positive QA pair and negative QA pair.

## I  EXPERIMENTAL RESULTS WITH DIFFERENT NUMBERS OF KGS

We put the results of SFT using different numbers of KGs, shown in Table 12. From the results, we can observe that the with the increase of the number of KGs, there is a slight improvement on the performances. However, from $N = 5$ to $N = 7$ there is a drop in EM score, which imply that more KGs may introduce noisy or wrong information, confusing the model.

Table 12: Experimental results of different number of KGs.

| Prompt Type | Method | TimeQA-Easy | | | | | TimeQA-Hard | | | | |
|---|---|---|---|---|---|---|---|---|---|---|---|
| | | R-1 | R-2 | R-L | BS | EM(%) | R-1 | R-2 | R-L | BS | EM(%) |
| W/ KGS (WHOLE-CONTEXT) | Qwen2.5-1.5B (SFT + $q$ + KGs (Faiss, $N = 3$)) | 0.236 | 0.135 | 0.236 | 0.458 | 0.17 | 0.221 | 0.152 | 0.220 | 0.433 | 0.20 |
| | Qwen2.5-1.5B (SFT + $q$ + KGs (Faiss, $N = 5$)) | 0.265 | 0.147 | 0.264 | 0.477 | 0.20 | 0.249 | 0.170 | 0.248 | 0.454 | 2.54 |
| | Qwen2.5-1.5B (SFT + $q$ + KGs (Faiss, $N = 7$)) | 0.285 | 0.159 | 0.284 | 0.490 | 0.07 | 0.267 | 0.180 | 0.267 | 0.462 | 0.39 |

## J  ABSTENTION PERFORMANCE COMPARISONS BETWEEN FULL-PARAMETER SFT AND LORA-SFT ON TIMEQA

We compare the abstention performance comparisons between full-parameter and LoRA SFT on TimeQA. For LoRA SFT training, we set LoRA rank $r = 32$, target modules are `q_proj` and `v_proj`, the dropout rate is $0.1$. We conduct SFT with `Qwen2.5-1.5B` on TimeQA-Easy and TimeQA-Hard, provided the context to the model, the results are shown in Table 13.

Table 13: Abstention performance comparisons between full-parameter SFT and LoRA-SFT on TimeQA

| | TimeQA-Easy | | | TimeQA-Hard | | |
|---|---|---|---|---|---|---|
| | TP | FP | FN | TP | FP | FN |
| Qwen2.5-1.5B-Instruct (SFT, Full-Param) | 91 | 309 | 231 | 120 | 342 | 199 |
| Qwen2.5-1.5B-Instruct (SFT, LoRA $r = 32$) | 71 | 437 | 251 | 141 | 682 | 178 |

Compared to full-parameter SFT, LoRA finetuning would increase False Positives (FP) on both tasks. However, LoRA SFT cannot address the "over-confidence" problem, which still holds true under the LoRA setting.

## K   RL Experiment with Different Data Distribution

In this section, we include another RL experiment with a balanced TimeQA-subset from TimeQA-Easy, to investigate the effect of the data distribution. As in Chen et al. (2021), for example, unanswerable questions only account for $12.4\%$ of the total data in the TimeQA-Easy training set, which means that the distribution of this data set is not balanced. Therefore, we create a subset where the proportion of unanswerable questions is $50\%$, by randomly selecting the same number of answerable data samples as the unanswerable data. Then we conduct the RL training with `Qwen2.5-1.5B-Instruct` under the same setting. The results are shown in Table 14.

**Analysis:**   Our experiments reveal that class imbalance critically affects RL optimization for answer abstention. In the natural distribution, the model largely ignores abstention because unanswerable cases are too rare to impact reward maximization. However, artificially balancing the dataset flips the problem: abstention becomes a trivial shortcut, which is a much simpler action space and can be optimized faster, leading the model to collapse into "always abstain." This indicates that achieving robust abstention requires not only carefully adjusting the reward design but also controlling the data distribution and training dynamics.

Table 14: RL experimental results comparisons between different data distributions on TimeQA-Easy.

| Proportion of Unanswerable Questions | TimeQA-Easy | | | |
|---|---|---|---|---|
| | TP | FP | FN | EM(%) |
| Original (12.4%) | 123 | 228 | 199 | 43.41 |
| Increased (50.0%) | 322 | 2544 | 0 | 11.24 |

## L   Constructing Abstained-version dataset from General QA Datasets

We use several general-purpose datasets to evaluate LLMs' abstention behavior in non-temporal settings, including **MMLU** and **HellaSwag**, which are multi-choice QA datasets, and **SQuAD-v2**. For MMLU and HellaSwag, we build upon the preprocessed versions provided by the **AbstainQA**[7] repository, and further modify them to simulate unanswerable scenarios.

The Massive Multitask Language Understanding (MMLU) benchmark (Hendrycks et al., 2020) evaluates a model's knowledge across 57 diverse subjects, including science, history, law, and medicine. Each question is multiple-choice with four options, testing both factual recall and reasoning ability. HellaSwag (Zellers et al., 2019) is a commonsense reasoning benchmark focused on sentence completion. Each instance presents a context followed by four possible continuations, only one of which is plausible. The task evaluates a model's ability to distinguish coherent continuations from adversarial distractors. SQuAD-v2 (Rajpurkar et al., 2016; 2018) is a reading comprehension dataset, consisting of questions posed by crowdworkers on a set of Wikipedia articles. Besides, it combines the 100000 answerable questions from SQuAD-1.1 with over 50000 unanswerable questions, adversarially written by crowdworkers to closely resemble answerable ones, which is suitable for our abstention experiments.

To ensure MMLU and HellaSwag datasets include unanswerable cases required for our experiments, for each dataset split (dev/test), we randomly select **12%** of the examples as the same as in TimeQA, and remove their correct answer choice from the option list. These examples are relabeled with a placeholder answer (`D`) and annotated as unanswerable. The remaining $88\%$ of examples are preserved, with a random incorrect option removed to ensure uniformity in choice count. All samples–modified and unmodified–are standardized to include exactly three options (`A`, `B`, `C`), with answer keys remapped accordingly. For SQuAD-v2 dataset, we randomly select 3000 questions for our experiment.

---

[7]`https://github.com/BunsenFeng/AbstainQA/tree/main`

This controlled perturbation enables us to systematically evaluate whether LLMs can correctly abstain, extending our analysis beyond temporally grounded QA tasks.

## M  ERROR ANALYSIS

In this section, we conduct the error analysis for the OOD QA experiments. Specifically, we analyze LLM's bebavior on SQuAD-v2, after RL trained on TimeQA-Easy.

---

**CASE STUDY #1**

**Question:** How are forces classified with regard to push and pull strengt? Context: Forces act in a particular direction and have sizes dependent upon how strong the push or pull is. Because of these characteristics, forces are classified as "vector quantities". This means that forces follow a different set of mathematical rules than physical quantities that do not have direction (denoted scalar quantities). For example, when determining what happens when two forces act on the same object, it is necessary to know both the magnitude and the direction of both forces to calculate the result. If both of these pieces of information are not known for each force, the situation is ambiguous. For example, if you know that two people are pulling on the same rope with known magnitudes of force but you do not know which direction either person is pulling, it is impossible to determine what the acceleration of the rope will be. The two people could be pulling against each other as in tug of war or the two people could be pulling in the same direction. In this simple one-dimensional example, without knowing the direction of the forces it is impossible to decide whether the net force is the result of adding the two force magnitudes or subtracting one from the other. Associating forces with vectors avoids such problems.
**LLM's Output:** Vector quantities
**Ground-Truth:** vector quantities

---

**CASE STUDY #2**

**Question:** As of what year were 10000 horsepower engines available? Context: In 1781 James Watt patented a steam engine that produced continuous rotary motion. Watt's ten-horsepower engines enabled a wide range of manufacturing machinery to be powered. The engines could be sited anywhere that water and coal or wood fuel could be obtained. By 1883, engines that could provide 10,000 hp had become feasible. The stationary steam engine was a key component of the Industrial Revolution, allowing factories to locate where water power was unavailable. The atmospheric engines of Newcomen and Watt were large compared to the amount of power they produced, but high pressure steam engines were light enough to be applied to vehicles such as traction engines and the railway locomotives.
**LLM's Output:** 1883
**Ground-Truth:** 1883

---

**CASE STUDY #3**

**Question:** What country no longer uses the Bowl of Hygieia as a symbol of pharmacy? Context: The two symbols most commonly associated with pharmacy in English-speaking countries are the mortar and pestle and the ℞ (recipere) character, which is often written as "Rx" in typed text. The show globe was also used until the early 20th century. Pharmacy organizations often use other symbols, such as the Bowl of Hygieia which is often used in the Netherlands, conical measures, and caduceuses in their logos. Other symbols are common in different countries: the green Greek cross in France, Argentina, the United Kingdom, Belgium, Ireland, Italy, Spain, and India, the increasingly rare Gaper in the Netherlands, and a red stylized letter A in Germany and Austria (from Apotheke, the German word for pharmacy, from the same Greek root as the English word 'apothecary').
**LLM's Output:** Netherlands
**Ground-Truth:** No Answer

As shown in Case#1 and Case#2, LLM answers correctly for reading comprehension, demonstrating that RL training in TimeQA-Easy shows that LLM learns to reason to solve some complex questions in OOD datasets. However, the model fails on abstention in Case#3, where the question does not have an answer.

