# OpenReview forum: "When Silence Is Golden: Can LLMs Learn to Abstain in Temporal QA and Beyond?"
_ICLR.cc/2026/Conference — ICLR 2026 Poster_

### Official Review · Reviewer_aoEZ · 2025-10-23

**Soundness:** 2
**Presentation:** 2
**Contribution:** 2
**Rating:** 2
**Confidence:** 4

**Summary:**

The paper studies abstention-aware temporal question answering with LLMs. It proposes a two-stage pipeline: (i) distill CoT rationales with GPT‑o1 and SFT the policy; (ii) continue training with GRPO using a simple, rule-based abstention-aware reward. The work also explores implicit (time-filtered sub-context, KG snippets) and explicit CoT reasoning signals. On TimeQA, the best model exceeds GPT‑4o with the original context by +3.46 and +5.80 EM, respectively. The paper further analyzes prompt designs, number of KGs, reward variants, and out‑of‑distribution generalization.

**Strengths:**

1. Timely focus on abstention + temporal reasoning.
2. Comparison across input settings (question only, full context, time-filtered sub-context, KGs), model scales, SFT vs RL, and prompt variants sheds insights into the domain.
3. The experimental setup is detailed nicely for reproduction.
4. Some interesting analysis is performed, including:
    4.1. SFT increases overconfidence
    4.2. Increasing unanswerable questions in training can collapse the model
    4.3. Impact of KG on abstention

**Weaknesses:**

1. **Lack of Benchmarks** - Results are confined to TimeQA. Other temporal reasoning sets (e.g., [1,2]) would better validate generality. The OOD experiments (Table 4, p. 9) focus on non‑temporal datasets and show very poor transfer after RL (e.g., TP -> 0 on RL+c), which underscores brittleness.

2. **Heavy reliance on GPT-o1 for CoT** - CoT collection use GPT‑o1. This raises questions about measuring knowledge distillation from larger models, rather than assessing the impact of suggested training and potential subtle leakage or bias from those systems.

3. **Training Methodology isn't novel** - The training stack (CoT‑SFT + GRPO) and rule‑based reward (Eq. 2) are standard training paradigms; the main novelty lies in the selected domain and results rather than new learning algorithms.

4. **Lack of LRMs** - Recent LRMs (eg, o4-mini, Gemini-2.5-Pro, or open-source LRMs -- DeepSeek-R1, Qwen-Thinking) have shown substantial reasoning improvements in temporal domains. Using them will shed more insights into state-of-the-art models and performance.

[1] Uddin, Md Nayem, et al. "UnSeenTimeQA: Time-Sensitive Question-Answering Beyond LLMs' Memorization." arXiv preprint arXiv:2407.03525 (2024).
[2] Fatemi, Bahare, et al. "Test of time: A benchmark for evaluating llms on temporal reasoning." arXiv preprint arXiv:2406.09170 (2024).

**Questions:**

See weaknesses above

---

> ### Author Response · Authors · 2025-11-18
> **Response to Reviewer aoEZ (1)**
>
> Thank you for your insightful review. We have addressed the points you raised as follows:
>
> ### **Q1: Results are confined to TimeQA. Other temporal reasoning sets (e.g., [1,2]) would better validate generality. The OOD experiments (Table 4, p. 9) focus on non‑temporal datasets and show very poor transfer after RL (e.g., TP -> 0 on RL+c), which underscores brittleness.**
>
> Thank you for the suggestion to use UnSeenTimeQA [1] and  Test of Time [2] for broader evaluation. However, unlike TimeQA, neither of these benchmarks includes explicit unanswerable or abstention-style questions. Because our work specifically studies abstention for temporal questions, these benchmarks do not align with our central goal. We chose to focus on abstention for temporal QA because of the model's limited ability to handle evolving factual knowledge and complex temporal logic [3], making it a rigorous setting for studying temporal abstention. Importantly, abstention is especially common and necessary in temporal question–answering scenarios: users frequently ask about events whose truth values change over time, outdated facts, or time-dependent attributes [4][5]. In such cases, models must decide whether the question is answerable given the available context, making temporal QA a natural real-world setting where abstention behavior is both common and essential. Besides, temporal QA datasets such as TimeQA provide clear answerable/unanswerable labels and analyzable context, enabling controlled evaluation that is difficult to achieve in general QA tasks.
>
> The observed drop in Table 4 is not merely brittleness, but rather a revealing finding: RL fine-tuning on temporal abstention sharpens decision boundaries specific to temporal tasks, which naturally reduces transfer to unrelated domains such as MCQ datasets that require a very different abstention logic. In other words, our results show that abstention behavior does not trivially generalize across task types, which we believe is an important and previously unreported insight, which requires further research.
>
> Finally, we note that our goal was not to propose a “universal abstention model”, but to conduct a systematic investigation of how implicit/external reasoning supervision and post-training paradigms influence abstention in temporal reasoning and OOD tasks. Extending our RL framework to broader abstention tasks is part of our planned future work.
>
> [3] Temporal knowledge question answering via abstract reasoning induction
>
> [4] It's High Time: A Survey of Temporal Information Retrieval and Question Answering
>
> [5] Time-aware language models as temporal knowledge bases
>
>
> ### **Q2: CoT collection use GPT‑o1. This raises questions about measuring knowledge distillation from larger models, rather than assessing the impact of suggested training and potential subtle leakage or bias from those systems.**
>
> Thank you for your comment. In our work, the CoT steps generated by GPT-o1 are not used for knowledge distillation or performance enhancement. We would like to clarify that for every question in the dataset we keep the original human-annotated response and also use it to train the models. The GPT-o1 is then used to generate CoT steps that can reason why the human-written response is correct. We then use the generated CoT to train reasoning models.
>
> Importantly, our goal is not to transfer knowledge from a larger model, but to **examine how different kinds of reasoning signals affect temporal abstention behavior.** Thus, even if GPT-o1 produces higher-quality and consistent CoT, this does not give the model advantage on the underlying task. CoT serves only as a kind of training signal for reasoning, not as ground-truth task labels.
>
> Furthermore, when collecting CoT steps, we provide the model with the full context from the dataset, which contains all the information required to answer each question. As a result, the GPT‑o1 is **not exposed to any additional external knowledge during CoT generation**, and there is no risk of knowledge leakage. Any stylistic biases in the generated CoT similarly do not affect our main findings. We have also manually checked the generated CoT samples and they appear high-quality.
>
> In addition, using strong LLMs to generate CoT supervision is a standard approach [6][7], and is widely accepted for controlled analysis.
>
> [6] Distilling Reasoning Capabilities into Smaller Language Models
>
> [7] MoDE-CoTD: Chain-of-Thought Distillation for Complex Reasoning Tasks with Mixture of Decoupled LoRA-Experts

---

> ### Author Response · Authors · 2025-11-18
> **Response to Reviewer aoEZ (2)**
>
> ### **Q3: The training stack (CoT‑SFT + GRPO) and rule‑based reward (Eq. 2) are standard training paradigms; the main novelty lies in the selected domain and results rather than new learning algorithms.**
>
> Our training stack (CoT-SFT + RL) and rule-based reward follow standard paradigms. However, the main contribution of our work lies not in proposing new learning algorithms, but in the systematic study of temporal abstention in LLMs. **Our work is the first to investigate both abstention and reasoning ability specifically in temporal QA, providing a comprehensive analysis of how different types of supervision, including implicit (context, sub-context, KG) vs. explicit (CoT), affect abstention behavior.**
>
> Our experiments reveal why SFT can induce overconfidence, and show that RL with abstention-aware rewards effectively improves True Positive rates on unanswerable questions in TimeQA-Easy. Together with OOD transfer analysis and detailed evaluation on temporal reasoning tasks, these constitute empirical findings that inform future design of LLMs with reliable abstention.
>
>
> ### **Q4: Recent LRMs (eg, o4-mini, Gemini-2.5-Pro, or open-source LRMs -- DeepSeek-R1, Qwen-Thinking) have shown substantial reasoning improvements in temporal domains. Using them will shed more insights into state-of-the-art models and performance.**
>
> Thanks for this advice. In order to address this, we conducted new experiments that we describe in the following. We have evaluated two LRMs (`o4-mini` and `Qwen3-4B-Thinking`) for more solid experiments, and the results are as follows:
>
> || TimeQA-Easy  | TimeQA-Hard |
> |-------| ------------- | ------------- |
> || R1 / R2 / RL / BS / EM(%) | R1 / R2 / RL / BS / EM(%) |
> |o4-mini| 0.634 / 0.519 / 0.632 / 0.711 / 44.14 | 0.531 / 0.439 / 0.530 / 0.639 / 37.12 |
> |Qwen3-4B-Thinking (2048 tokens for Thinking)| 0.328 / 0.271 / 0.277 / 0.371 / 24.56 | 0.278 / 0.221 / 0.277 / 0.333 / 21.51|
>
> The used prompt in this experiment was:
> > You are a top expert at answering questions about time. Please think step by step, and give the answer between \<answer\> and \</answer\> according to the information provided. \
> If there is no answer towards the question in the context, please respond \"No Answer\". \
> Do not search the internet and website.
>
>
> Based on the results, we can observe that reasoning models can outperform non-reasoning models (i.e., `GPT-4o` and `Qwen2.5-7B-Instruct`). It is also worth noting that our 1.5B reasoning model can achieve the comparable performance with o4-mini. This addresses the issue you raised and will be a useful insight to include in the paper. We will add these new results in the revised version of our paper.

---

> > ### Comment · Reviewer_aoEZ · 2025-11-23
> >
> > Thank you for the response. While most of my concerns are addressed, I am still struggling to grasp the main contribution. From my understanding, the goal of the paper is to analyze how current methods hold up when training on temporal questions, specifically when the model should abstain from answering. Neither the problem setting nor the methods are novel. While the current training methods do help, as documented by the authors, they observe a significant drop in general capabilities on OOD benchmarks. Any insights as to why this happens and, more importantly, how this can be overcome, will boost the paper's contribution significantly.

---

> > > ### Author Response · Authors · 2025-11-23
> > > **Response to Reviewer aoEZ**
> > >
> > > We appreciate the reviewer’s feedback and we are glad to hear that our response answered your concerns. Our paper is the first work which addresses LLM abstention on time-sensitive QA. We studied several approaches from knowledge graphs to reinforcement learning to achieve this goal. Therefore, we would like to argue that our problem setting is novel and we have explored a variety of approaches and showed what does work and what doesn’t. As you know the number of very influential LLM training methods introduced by the community, is limited to a few or less every year, so we hope that the fact that we do not introduce a novel training method would not be a negative point of our paper.
> > >
> > > Regarding your question related to insight on OOD performance, we observe from Table 4 in our paper that temporal-uncertainty training using SFT + RL disproportionately shifts the model’s calibration behaviour, while mere use of SFT improves the model performance. This experiment is presented in Table 4 in the paper. Based on this and other papers from the literature [1], its a general understanding that while RL training significantly improves a model’s alignment with a certain desired behaviour, it can also force it to digress from its usual behaviour on other tasks/domains when compute or data is small. More specifically, the paper titled “SFT Memorizes, RL Generalizes: A Comparative Study of Foundation Model Post-training” [1], demonstrates that for RL approaches, with growing number of data and compute (i.e. G-FLOPS as shown in the paper) the model can generalize on OOD tasks. Based on this discussion and the findings in [1] we believe that in order to achieve a robust generalization, one needs a larger dataset and more G-FLOPS.
> > >
> > > We would like to emphasize that our problem definition is different from temporal reasoning, as we are only focused on LLM abstention on time-sensitive QA. We will add these points to the paper in a discussion section. We would highly appreciate it if our rebuttal could be positively reflected in your final evaluation scores.
> > >
> > > Reference
> > > 1. Chu, T., Zhai, Y., Yang, J., Tong, S., Xie, S., Schuurmans, D., Le, Q.V., Levine, S. and Ma, Y., 2025. Sft memorizes, rl generalizes: A comparative study of foundation model post-training. arXiv preprint arXiv:2501.17161.

---

> > > > ### Author Response · Authors · 2025-11-26
> > > > **Response to Reviewer aoEZ (Update)**
> > > >
> > > > Thank you again for taking the time to provide us feedback. We would like to inform you that based on your feedback and other reviewers' feedback, we modified the paper and updated the paper.
> > > >
> > > > Reviewer eqVi also read our discussions with you and suggested improvements of the paper story-line which we already did. We believe that the updated story-line makes our contributions more clear and can potentially address the points you raised. Namely, we emphasized that our focus is on Temporal reasoning with abstention where an LLM learns when to refuse to answer (e.g. cases where a question is unanswerable).
> > > >
> > > > We also added the experiment results presented above along with other discussions with you to the paper. We would highly appreciate it if you could check these new modifications (which are marked with red font) and let us know if any further changes are needed. We would be happy to address any further points you may raise.

---

> > > > > ### Comment · Reviewer_aoEZ · 2025-11-26
> > > > >
> > > > > Thank you for making efforts during the rebuttal period. I agree with reviewer eqVi that the paper reads better now and is in a better position, but I do not agree with the statement *"Our paper is the first work which addresses LLM abstention on time-sensitive QA"*. As noted by the authors, the abstention tasks are introduced in TimeQA, and the problem domain isn't novel.
> > > > >
> > > > > Moreover, a recent work mentioned by the reviewer eqVi [1], studies abstention capabilities using RL, which is similar to the study done in this paper. That being said, authors have made clear distinctions from the mentioned work and, to the best of my knowledge, are the first ones to study existing methods for abstention in Temporal QA. I believe these insights might be useful to the community. I have increased my score to 4.
> > > > >
> > > > > [1]: Song, L., Shi, T., & Zhao, J. (2025). The hallucination tax of reinforcement finetuning. arXiv preprint arXiv:2505.13988.

---

> > > > > > ### Author Response · Authors · 2025-11-28
> > > > > > **Response to reviewer aoEZ**
> > > > > >
> > > > > > Thank you for raising your scores and for your invaluable feedback. We truly appreciate your willingness to work with us to clarify and improve the presentation of our paper.
> > > > > >
> > > > > > Following your suggestions, we have revised our stated contribution to clarify that we are the first to systematically study and compare how different information types and training techniques affect temporal reasoning with abstention behavior in LLMs. We also cited the paper “The hallucination tax of reinforcement finetuning” in the related work and compared the differences. These changes have been incorporated into the PDF, which is available for your review.
> > > > > >
> > > > > > Finally, we would like to present our interesting findings and novel insights from this study, which to the best of our knowledge, are presented for the first time on abstention tasks in temporal QA:
> > > > > >
> > > > > > - Abstention in temporal QA tasks is a learnable skill which could be guided with RL.
> > > > > > - Starting RL from SFT with CoTs equips smaller models with core reasoning with abstention skills, enabling them to rival or surpass larger closed-source models like GPT-4o.
> > > > > > - Contexts and KGs provide limited improvement compared with CoT in the task of abstention in temporal QA.
> > > > > > - SFT can induce overconfident behavior in LLMs; RL can enhance reasoning with abstention ability, but the risk of overconfidence still exists.
> > > > > > - Abstention ability in temporal QA tasks is difficult to generalize to out-of-distribution domains, highlighting the need for broader training coverage and more careful model design.
> > > > > >
> > > > > > Thank you again for taking the time to provide us feedback. We are open to any further suggestions that you may have and we are more than willing to make any changes to the paper or address any further issues.

---

### Official Review · Reviewer_Ght9 · 2025-10-28

**Soundness:** 3
**Presentation:** 3
**Contribution:** 3
**Rating:** 8
**Confidence:** 4

**Summary:**

This paper studies the important question of how to teach large language models (LLMs) the skill of abstention: not answering a question. This work focuses on questions involving a temporal dimension. The authors explore various forms of SFT and RL (using GRPO) to induce abstention. The authors show training a model with chain-of-thought supervised finetuning followed by reinforcement learning can boost abstention. The authors highlight that generalizing outside of the TimeQA benchmark to out of domain benchmarks such as MMLU is still challenging.

**Strengths:**

The authors tackle the important open problem of the best approach to teach models the skill of abstention for temporal questions. The authors cover a reasonable set of closed and open models as well as explore various setups for inducing abstention, including various approaches to including context. The authors make reasonable choices in terms of post-training methods (GRPO, SFT) and adapt them for abstention. I commend the authors on the perspective that abstention is a learnable skill and the thorough exploration in post-training approaches to induce it.

I appreciate the authors were careful about evaluation by including both correct abstention and over-abstention in the experiments. I also appreciate the evaluation of both in-domain TimeQA as well as other out-of-domain benchmarks to assess generalization. I also appreciate the authors' proactive inclusion of limitations such as model size.

The findings are quite interesting and offer a practical recipe for inducing abstention for temporal questions. The finding regarding the importance of data mix (answerable versus unaswerable) is also quite neat! The authors also highlight the important open problem of teaching LLMs the skill of abstention more generally, as well as the limitations of only teaching abstention using SFT.

**Weaknesses:**

The authors offer some nice findings comparing post-training approaches for abstention, including the lack of success of some approaches (SFT). One aspect that could be improved here is some more intuition regarding why some setups work better than others. There is a growing body of literature along the lines of https://arxiv.org/abs/2501.17161 which explains memorization and generalization learning dynamics of post-training approaches that can be used to better contextualize this works' findings.

The hyperparameters and exact setups used can play a large factor here. The authors' present claims in quite a general manner, not sufficiently accounting for the limited setup used to justify them. For example, broad claims about SFT versus RL are supported only with a single method (GRPO) or limited hyperparameter selection choices for SFT. I'd also be curious to see whether the result in line 397 holds with LoRA, which as been shown to reduce overfitting.

The experiments regarding generalization are quite interesting. I imagine most common post-training setups will include other data for alignment. How would this skill of abstention interact with the standard post-training pipeline aimed at aligning models?

While I believe temporal questions are certainly important, I believe the authors could do a better job setting up why it's worth focusing only on temporal questions. Adding more context about why temporal questions are particularly important or worth focusing on solely would help to better frame the contribution.

**Questions:**

- While temporal questions are certainly important, have the authors considered whether this approach would generalize to other types of unanswerable questions? It's not necessary to run these additional experiments for the scope of this paper, but it would be useful to discuss in the context of future work.
- How are the hyperparameters in lines 244-246 selected? Is there precedent or justification in prior work or was a sweep conducted?

---

> ### Author Response · Authors · 2025-11-18
> **Response to Reviewer Ght9 (1)**
>
> Thank you for your insightful review. We have addressed the points you raised as follows:
>
> ### **Q1: One aspect that could be improved here is some more intuition regarding why some setups work better than others. There is a growing body of literature along the lines of https://arxiv.org/abs/2501.17161 which explains memorization and generalization learning dynamics of post-training approaches that can be used to better contextualize this works' findings.**
>
> Thanks for this useful suggestion, and we agree with the reviewer. This paper aligns well with our observations in temporal-abstention: SFT tends to memorize surface patterns from the training data and thus becomes over-confident, while RL optimizes outcome-level rewards and therefore generalizes better, and SFT is necessary for RL training as a cold-start.
> We will cite this paper in our final version, and add more discussion using the additional page.
>
>
> ### **Q2: For example, broad claims about SFT versus RL are supported only with a single method (GRPO) or limited hyperparameter selection choices for SFT. I'd also be curious to see whether the result in line 397 holds with LoRA, which as been shown to reduce overfitting.**
>
> Thanks for this insightful comment. In order to address this comment and understand the interaction pattern between SFT vs. RL we conducted new experiments which we will explain in the following:
> + The reason why we choose GRPO is that it is efficient without the need for a reward model and also one of the most representative methods for RL training.
> + For hyperparameter selection, these are some default choices for training, such as the learning rate and weight decay. We acknowledge the importance of more hyperparameter experiments to make the paper more solid, and we will add more ablation experiments for the final version of the paper in the appendix.
> + For LoRA SFT training, we choose LoRA rank $r=32$, target modules are `q_proj` and `v_proj`, the dropout rate is $0.1$. We conduct SFT with `Qwen2.5-1.5B` on TimeQA-Easy and TimeQA-Hard, provided the context to the model, the results are as follows:
>
> | TimeQA-Easy  | TimeQA-Hard |
> | ------------- | ------------- |
> | R1 / R2 / RL / BS / EM(%) |  R1 / R2 / RL /BS  /EM(%)   |
> |  0.321 / 0.184 / 0.316 / 0.519 / 1.29 | 0.246 / 0.153 / 0.243 / 0.467 / 2.37 |
>
> The abstention performances are as follows:
>
> || TimeQA-Easy  | TimeQA-Hard |
> |-------| ------------- | ------------- |
> || TP / FP / FN |  TP / FP / FN   |
> |Qwen2.5-1.5B-Instruct (SFT, Full-Param)| 91 / 309 / 231  | 120 / 342 / 199 |
> |Qwen2.5-1.5B-Instruct (SFT, LoRA r=32)| 71 / 437 / 251  | 141 / 682 / 178 |
>
> Compared to full-parameter SFT, LoRA finetuning would increase False Positives (FP) on both tasks. However, LoRA SFT cannot address the "over-confidence" problem, which still holds true under the LoRA setting. We will add this experiment in the revised version of our paper.
>
> ### **Q3: I imagine most common post-training setups will include other data for alignment. How would this skill of abstention interact with the standard post-training pipeline aimed at aligning models?**
>
> Thank you for this insightful question. Modern post-training pipelines (e.g., SFT → RLHF/DPO/GRPO) typically include diverse alignment data, but they do not reliably teach models epistemic abstention (i.e., recognizing when an answer is unknowable). In fact, our experiments show that standard SFT tends to make models more confident and reduces abstention even when abstention examples are present (Section. 6.3). This suggests that generic alignment signals do not automatically produce the behavior we study.
>
> Our abstention-aware rewards can be viewed as an additional, modular alignment signal that can be inserted into existing pipelines. The reward encourages honesty on unanswerable cases and penalizes hallucinations, similar in spirit to how safety RLHF introduces domain-specific preferences. Thus, our method is compatible with standard post-training; it simply targets a skill that is otherwise underrepresented in typical alignment data.
>
> Finally, we note that abstention interacts nontrivially with other alignment objectives: SFT increases overconfidence, and RL may reduce abstention on harder queries without an explicit abstention reward. This indicates that abstention should be treated as a distinct alignment objective rather than something expected to emerge implicitly. We will add this analysis into the discussion of the final version of our paper.

---

> ### Author Response · Authors · 2025-11-18
> **Response to Reviewer Ght9 (2)**
>
> ### **Q4: I believe the authors could do a better job setting up why it's worth focusing only on temporal questions. Adding more context about why temporal questions are particularly important or worth focusing on solely would help to better frame the contribution.**
>
> Thanks so much for your suggestions. We chose temporal QA because it introduces inherent temporal inconsistency and ambiguity, making it a rigorous setting for studying abstention. Importantly, abstention is especially common and necessary in temporal question–answering scenarios: users frequently ask about events whose truth values change over time, outdated facts, or time-dependent attributes. In such cases, models must decide whether the question is answerable given the available context, making temporal QA a natural real-world setting where abstention behavior is both common and essential. Besides, temporal QA datasets such as TimeQA provide clear answerable/unanswerable labels and analyzable context, enabling controlled evaluation that is difficult to achieve in general QA tasks.
>
> We will strengthen the introduction to motivate temporal reasoning for abstention.
>
> ### **Q5: While temporal questions are certainly important, have the authors considered whether this approach would generalize to other types of unanswerable questions? It's not necessary to run these additional experiments for the scope of this paper, but it would be useful to discuss in the context of future work.**
>
> T​hank you for the thoughtful question. Yes, our approach is designed to generalize and we expect it to transfer to other categories of unanswerable questions beyond temporal QA. In this paper, we focus on temporal questions because they naturally contain frequent ambiguity, temporal inconsistency, and context-dependent answerability, providing a controlled yet challenging environment to study abstention.
>
> Although we do not include experiments on other types of unanswerable questions due to the scope in this work, we agree that evaluating broader generalization is important. We plan to extend our framework to domains such as multi-hop reasoning and ambiguous fact-based QA in the future work, as these settings also require reliable abstention behavior.
>
>
> ### **Q6: How are the hyperparameters in lines 244-246 selected? Is there precedent or justification in prior work or was a sweep conducted?**
>
> Thank you for pointing this out. For the hyperparameters in lines 244–246, we followed commonly used default settings from prior post-training work rather than conducting an extensive sweep. Our goal in this paper is to analyze and compare the  behaviors of different post-training paradigms under a consistent default configuration.
>
> We agree that a broader hyperparameter sweep could further strengthen the conclusions, and we plan to explore more systematic tuning, especially for SFT and RL in future work.

---

### Official Review · Reviewer_eqVi · 2025-10-31

**Soundness:** 2
**Presentation:** 1
**Contribution:** 2
**Rating:** 2
**Confidence:** 3

**Summary:**

This paper studies how RL can teach language models to abstain from answering when uncertain, particularly in temporal question answering. The authors introduce an RL framework with abstention-aware rewards where models are rewarded for saying "no answer" on unanswerable questions. Their method outperforms both GPT-4o and SFT baselines, showing higher accuracy and better handling of unanswerable questions. Results are supported by numerous ablations over prompt configurations, hyper-parameter tuning and generalization.

**Strengths:**

- **Relevance and Importance**: The paper addresses an important and timely problem. Current models struggle to abstain and the proposed RL technique is simple and effective.

- **Strong Results**: The results, although surprising, are strong. RL significantly beats SFT, even for SFT models of larger sizes.

**Weaknesses:**

- **Poor Structure and Presentation**: The paper’s organization lacks coherence. Sections jump between unrelated topics (e.g., implicit reasoning, KG extraction, RL training) without clear motivation or integration into the main story. Figures and experiments are presented out of logical order, reducing readability.

- **Weak Experimental Design**: Some experiments feel arbitrary or poorly motivated. Dataset choices, baselines, and prompt configurations are insufficiently justified, and several results are unintuitive.

- **Limited Applicability**: This approach assumes access to datasets with unanswerable questions, which is generally not applicable. The proposed approach also seems very targeted to TemporalQA, which further limits applicability.

**Questions:**

- **Classifier Baseline:** Can the authors add a classifier baseline that predicts if a question is answerable? This could be combined with any model that always generates an answer, and is a simple post-hoc baseline that can compliment models that struggle to abstain.

- **Existing Work:** There is some prior work [1] training models to abstain using RL on unanswerable questions. Can the authors discuss novelty compared to this (and possibly more) existing works. It is okay if they were concurrent works.

- **SFT Data:** Does the SFT data also include unanswerable questions? If so, how are these distributed? They should ideally have the same proportion of unanswerable questions as the RL training (where data ratio was so critical).

- **SFT performance:** Why is SFT performance so poor relative to RL? The SFT dataset appears to have only 1K examples compared to 20K for RL. Could this explain the gap? Note that I am not surprised by the fact that RL beats SFT, but rather by the margin of defeat. A 1.5B model beating a 8B model this significantly suggests that the SFT pipeline has issues.

- **Frontier Model evaluations:** For frontier model evaluations, what prompts were used? Were models explicitly told they could abstain? I think the best prompt to use for these evaluations is exactly the training prompt in Table 8 (except the Qwen/Alibaba text).

- **Implicit Reasoning:** What is the importance of implicit reasoning, such as temporal reasoning or KG extraction, in the story of this paper? In particular, the best models seem to perform well even without these augmentations—what value do these methods add? If the focus of this paper is RL, then the implicit reasoning methods should be presented as baselines.

- **Generalization**: The OOD generalization tasks differ substantially. In TimeQA, abstention is due to lack of information; in MCQ datasets, abstention reflects model inability. Are these two forms of abstention comparable? Why not evaluate on AbstentionBench, which contains ambiguous/unanswerable tasks and seems more aligned with the paper’s goals?

- **Focus on TemporalQA**: The emphasis on temporal QA is unclear. Why is this chosen as the focus, when the RL framework could generalize to other question-answering tasks which require abstention as well (multi-hop reasoning for example) ?

- **Qwen 2.5-7B Results**: The results of this model on TimeQA-Hard are surprising. EM Accuracy is highest when only given the question. Context seems to reduce performance. Why is this happening?

- Section 3.1 should clearly specify that some questions are explicitly unanswerable.

- The figures are misordered and disrupt reading flow. For instance, results for Figure 4 precede those for Figure 3, and Figure 5 appears beside unrelated text (Experiment 6.2).

[1]: Song, L., Shi, T., & Zhao, J. (2025). The hallucination tax of reinforcement finetuning. arXiv preprint arXiv:2505.13988.

---

> ### Author Response · Authors · 2025-11-18
> **Response to Reviewer eqVi (1)**
>
> Thanks for the insightful comments and feedback. We have addressed all the issues you raised by also running additional experiments. Please see our responses below:
>
>  ### **Q1: A classifier baseline to predict if a question is answerable or not.**
>
> Thanks for this helpful suggestion. We trained a classifier using `Qwen2.5-1.5B-Instruct`, with a classification head to classify if the question is answerable or not (i.e., 0 for unanswerable and 1 for answerable questions). If we got the 0 label during evaluation inference, we set the LLM’s answer to ``No Answer”; otherwise, we keep the original generated answer from LLM. We test `Qwen2.5-1.5B-instruct` classifier baseline with `Qwen2.5-7B-Instruct` as the LLM of choice. As input, we provide `question + context` to the model. The number of training epochs is determined at 2 beyond which the loss does not decrease. The evaluation results on the two datasets of TimeQA-Easy and TimeQA-Hard are as follows:
>
>
>
> | TimeQA-Easy  | TimeQA-Hard |
> | ------------- | -------------|
> | R1 / R2 / RL / BS / EM(%) |  R1 / R2 / RL /BS  /EM(%)   |
> | 0.159/ 0.126 / 0.158 / 0.387 / 11.41   | 0.138 / 0.128 / 0.137 / 0.357 / 12.40  |
>
>
> The baseline shows that the classifier does not work well for predicting whether a temporal question is answerable or not. This confirms that our approach is very useful and effective for the abstention task. We will add this baseline in the revised version.
>
> ### **Q2: Discuss the novelty compared to “The hallucination tax of reinforcement finetuning. arXiv preprint arXiv:2505.13988.”**
>
> Thank you for pointing out this related work. We note that this is a concurrent study, and is complementary to our setting in several important ways. First, their focus is on synthetic unanswerable math problems, whereas our work targets general temporal reasoning, which requires reasoning over time-scoped facts, detecting temporal contradictions.
>
> Second, the RL objectives differ significantly. Song et al. design verification-based rewards including correctness and explicit abstention incentives, tailored specifically to math verification tasks. In contrast, our reward design
> optimizes both abstention and reasoning ability of LLM, with Exact-Match and Rouge score.
>
> Third, their method depends on constructing a dedicated synthetic unanswerable dataset for finetuning, while our model learns abstention behavior directly through RL from naturally occurring unanswerable temporal questions.
>
> In conclusion, this work and ours study abstention from different perspectives and domains. We will cite the mentioned paper in the related work, however, to our knowledge, our work is the first to systematically investigate RL-driven abstention in temporal QA.
>
> ### **Q3: Does the SFT data also include unanswerable questions? If so, how are these distributed? They should ideally have the same proportion of unanswerable questions as the RL training (where data ratio was so critical).**
>
> Thank you for the question. We would like to clarify that our SFT data (both vanilla SFT and SFT as cold-start for RL) includes unanswerable questions as the original datasets.
> + Vanilla SFT: This experiment directly uses the original TimeQA data (answerable + unanswerable) with their natural proportions.
>
> + CoT-SFT for RL cold start: The CoT-augmented SFT set also keeps the same answerable/unanswerable ratio as the original dataset. We only add reasoning chain-of-thought steps for SFT training.

---

> ### Author Response · Authors · 2025-11-18
> **Response to Reviewer eqVi (2)**
>
> ### **Q4: Why is SFT performance so poor relative to RL? The SFT dataset appears to have only 1K examples compared to 20K for RL. Could this explain the gap? Note that I am not surprised by the fact that RL beats SFT, but rather by the margin of defeat. A 1.5B model beating a 8B model this significantly suggests that the SFT pipeline has issues.**
>
> Thank you for the question and here is a detailed answer:
> + The SFT experiments do not use only the 1K CoT samples, rather it uses the entire dataset. Both vanilla SFT and RL are trained on the full TimeQA training set, which contains tens of thousands of examples. The smaller CoT subset (about ~2K for Easy and ~1K for Hard) is used only for the cold-start SFT stage prior to RL, not as the main SFT dataset.
> + Regarding the performance gap, the SFT pipeline is functioning as expected. In fact, SFT significantly improves over direct inference (INF) on ROUGE and EM, confirming that the SFT model is learning meaningful patterns. For example, the SFT trained `Qwen2.5-1.5B-Instruct` model has a much higher ROUGE score on TimeQA-Easy than direct inference using  `Qwen2.5-1.5B-Instruct` model. However, compared to SFT, the RL-trained model has better abstention and reasoning abilities, which indicates the superiority of the RL method.
>
>
> ### **Q5: For frontier model evaluations, what prompts were used? Were models explicitly told they could abstain? I think the best prompt to use for these evaluations is exactly the training prompt in Table 8 (except the Qwen/Alibaba text).**
>
> Thank you for raising this important point. For all frontier model evaluations, we use the prompt as follows:
> > You are a top expert at answering questions about time. You have to answer the questions without any explanations according to the context. \
> If there is no answer towards the question in the context, please respond \"no answer\". \
> Do not search the website.
>
> The prompt explicitly instructs all models that they may respond with ‘no answer’. The instruction “Do not search the website” ensures that frontier models do not use online tools, retrieval, or proprietary browsing capabilities, which are capabilities that open-source models do not possess. This guarantees that all systems operate purely on internal model knowledge and provided context, creating a fair, controlled comparison.
> We will include all evaluation prompts in the appendix for full transparency.
>
> ### **Q6: What is the importance of implicit reasoning, such as temporal reasoning or KG extraction, in the story of this paper? In particular, the best models seem to perform well even without these augmentations—what value do these methods add? If the focus of this paper is RL, then the implicit reasoning methods should be presented as baselines.**
>
> Thanks for this question. In our work, the primary goal is not merely to propose the most effective model, but to **provide a comprehensive analysis of how different information sources and training techniques affect LLMs’ performance on both temporal reasoning and abstention tasks**. We categorize these information sources into two types: implicit (context, sub-context, knowledge graphs) and explicit(chain-of-thought supervision). Through extensive experiments, we find that implicit reasoning methods, such as extracting additional context or KG information, do not significantly improve abstention ability. Importantly, ruling out these ineffective approaches is itself a valuable contribution, as it clarifies which strategies are not helpful for temporal abstention. Besides this, we observe that reinforcement learning setups are effective. Therefore, implicit reasoning methods should not be viewed as baselines for RL, and they are instead a key part of our analytical investigation into the factors influencing temporal abstention.

---

> ### Author Response · Authors · 2025-11-18
> **Response to Reviewer eqVi (3)**
>
> ### **Q7: The OOD generalization tasks differ substantially. In TimeQA, abstention is due to lack of information; in MCQ datasets, abstention reflects model inability. Are these two forms of abstention comparable? Why not evaluate on AbstentionBench, which contains ambiguous/unanswerable tasks and seems more aligned with the paper’s goals?**
>
> Thank you for this question. We would like to clarify that in TimeQA, abstention is not simply due to a lack of information. In our experiments, we provide models with various forms of information in implicit reasoning, including context, sub-context with temporal information, and knowledge graphs, which collectively contain the information necessary to answer the questions. If the model’s reasoning ability is sufficient, it can infer whether to abstain from these sources and infer the correct answer. Therefore, in TimeQA, failures to abstain correctly are primarily due to limitations in the model’s reasoning and judgment abilities.
>
> Regarding AbstentionBench, this benchmark was proposed very recently (June 2025), after we had already finalized our study, which is why it was not included. We will to evaluate our methods on AbstentionBench in the revised version as you suggested to further complement our analysis of OOD generalization.
>
> ### **Q8: The emphasis on temporal QA is unclear. Why is this chosen as the focus, when the RL framework could generalize to other question-answering tasks which require abstention as well (multi-hop reasoning for example)?**
>
> There are several reasons why we emphasize on temporal QA:
>
> + Temporal QA is an especially challenging domain because it requires models to reason over evolving events and shifting timelines[1], where facts often become ambiguous or contradictory over time (e.g., a marriage ending in divorce) [2, 3]. This inherent complexity makes the abstention decision significantly more difficult than in static QA. In fact, our results on GPT-4o clearly show that GPT-4o struggles with temporal QA abstention and is inferior to our best model.
>
> + TimeQA dataset provides unanswerable questions and clear answerable/unanswerable labels along with analyzable context, making them particularly suitable for studying and evaluating abstention behavior.
>
> + While our RL framework is general and could be applied to other abstention-requiring tasks such as multi-hop reasoning, we focus on temporal QA in this work because temporal reasoning presents greater challenges than standard QA, making it a more rigorous testbed for analyzing abstention behavior. We plan to extend our framework to more general tasks in future work.
>
> [1] Chen Z, Li D, Zhao X, et al. Temporal knowledge question answering via abstract reasoning induction[C]//Proceedings of the 62nd Annual Meeting of the Association for Computational Linguistics (Volume 1: Long Papers). 2024: 4872-4889.
>
> [2] Piryani B, Abdullah A, Mozafari J, et al. It's High Time: A Survey of Temporal Information Retrieval and Question Answering[J]. arXiv preprint arXiv:2505.20243, 2025.
>
> [3] Dhingra B, Cole J R, Eisenschlos J M, et al. Time-aware language models as temporal knowledge bases[J]. Transactions of the Association for Computational Linguistics, 2022, 10: 257-273.
>
> ### **Q9: The results of this model on TimeQA-Hard are surprising. EM Accuracy is highest when only given the question. Context seems to reduce performance. Why is this happening?**
>
> This is indeed an interesting phenomenon. For GPT-3.5 and GPT-4o, adding the context generally improves both Rouge and EM scores, as expected. However, for the Qwen2.5 series models, we observe a drop in performance when context is provided. A possible explanation is that these models may over-rely on the provided context, which can sometimes contain distracting information, especially in the TimeQA-Hard setting where questions are more complex and temporally nuanced.
>
>  In fact, recent work shows that long contexts can distract an LLM if not very relevant to a task. A recent study focused on RAG models [4] introduces a method named focus mode where they only keep top similar sentences to the question in the input context. They show that this kind of context pruning outperforms baseline. In such cases, the model may focus on irrelevant parts of the context, leading to suboptimal reasoning and lower EM scores.
>
> Another contributing factor could be domain mismatch or insufficient pretraining on complex temporal reasoning for the Qwen2.5 series, making them less capable of effectively integrating additional contextual information. We will add further discussion and analysis in the paper to better understand this behavior.
>
> [4] Li S, Stenzel L, Eickhoff C, et al. Enhancing retrieval-augmented generation: a study of best practices[J]. arXiv preprint arXiv:2501.07391, 2025.

---

> ### Author Response · Authors · 2025-11-18
> **Response to Reviewer eqVi (4)**
>
> ### **Q10: Section 3.1 should clearly specify that some questions are explicitly unanswerable.**
>
> Thanks for pointing this out! We will modify the paper in the final version and specify it clearly.
>
> ### **Q11: The figures are misordered and disrupt reading flow. For instance, results for Figure 4 precede those for Figure 3, and Figure 5 appears beside unrelated text (Experiment 6.2).**
>
> Thanks for this comment. The main reason why the figures are a bit misordered is due to the limited space of the paper, as we tried our best to fit them into the paper. We will fix this issue in the final version using the additional page.

---

> > ### Comment · Reviewer_eqVi · 2025-11-24
> > **Response to rebuttal**
> >
> > Thank you, a significant number of my concerns have been address. After reading through reviewer aoEZ's review, I do agree that the main contribution and story needs to be made clearer. If the main focus of this work is on understanding different approaches to abstaining in temporal QA, then it needs to be brought to light more. For example, the abstract mainly focuses on reinforcement learning, and does not even mention the KG extraction/ other extraction methods which RL is compared to. Quoting the author's rebuttal:
> >
> > "*the primary goal is not merely to propose the most effective model, but to provide a comprehensive analysis of how different information sources and training techniques affect LLMs’ performance on both temporal reasoning and abstention tasks. We categorize these information sources into two types: implicit (context, sub-context, knowledge graphs) and explicit(chain-of-thought supervision). Through extensive experiments, we find that implicit reasoning methods, such as extracting additional context or KG information, do not significantly improve abstention ability. Importantly, ruling out these ineffective approaches is itself a valuable contribution, as it clarifies which strategies are not helpful for temporal abstention.*"
> >
> > This is not made clear through the abstract, or even the introduction, which made me think the primary contribution was a RL method for temporalQA. I have raised my score, but will only raise it further if the writing and story are improved:
> >
> > 1. The main story is "evaluating LLMs on temporal QA", the introduction/abstract/results/discussion should bring this into focus.
> > 2. As stated in my original review, the figures are misordered and disrupt reading flow. Please reorder the paper to ensure that the results section flows smoothly. Table 1 is also extremely uninterpretable, there are too many things going on at once. Maybe there is a way to break this table up, or move some of the results to appendix.
> >
> >  It is possible to update the paper during the rebuttal, so these changes can be made.

---

> > > ### Author Response · Authors · 2025-11-26
> > > **Response to Reviewer eqVi**
> > >
> > > Thank you for your insightful and constructive feedback. We are glad to hear that our rebuttal addressed a significant number of your concerns. Following your feedback, we modified the paper to present a clearer story-line bringing the main focus of the paper (i.e. Temporal Reasoning with Abstention) to light.
> > >
> > > As you can see in the new version of the paper, we modified the abstract, introduction, results, and conclusion to address your comment and achieve an improved story-line. Additionally, we fixed the issue with the misordered figures to make sure the paper reads fluently and with a logical flow. Furthermore, we broke down Table 1 into two tables (i.e. Table 1 and Table 2) and we believe that this has improved clarity of the presentation. Finally, we added to the paper all the experiments' results presented in the rebuttal.
> > >
> > > We believe that this rebuttal and your invaluable feedback have significantly improved our paper. We hope that these modifications would be reflected positively in your final assessment scores. We would be happy to address any further concerns or make any other modifications to the paper as you suggest.

---

> > > > ### Comment · Reviewer_eqVi · 2025-11-26
> > > > **Response to rebuttal**
> > > >
> > > > Thank you, the paper reads much better now. I have raised my score again.

---

> > > > > ### Author Response · Authors · 2025-11-28
> > > > > **Response to reviewer eqVi**
> > > > >
> > > > > Thank you very much for your constructive review. The discussions we had with you resulted in a significant improvement of our paper. We are open to any further suggestions or comments you may have.

---

### Author Response · Authors · 2025-12-01
**Summary of Reviews and Discussions**

Dear AC,

Thank you for taking the time to review our paper. We highly appreciate your efforts as well as the insightful and constructive discussions we had with the three reviewers. **The two reviewers with the lowest scores, both increased their scores before the glitch incident.** They did so based on our additional experiments that they asked for and the improved storyline of the paper, which clarifies our main contributions. These changes that we made to the revised PDF, addressed the reviewers’ concerns and resulted in them increasing their scores. **The average rating score of our paper before the glitch happened was 6.**

**We recap the main issues raised by the reviewers that we addressed:**
- [Motivations and main contributions] need for clarifying motivations and main contributions  and improving presentation (eqVi), clarifying distinctions with previous work (eqVi; aoEZ).
- [Additional experiments done] included a classifier baseline (eqVi), LoRA-SFT (Ght9), and evaluating additional large reasoning models (aoEZ).
- [Required experiment details, and future work discussion] included experiments’ details (eqVi; Ght9; aoEZ), and discussion of possible future work (Ght9).

**We summarize our responses and reviewer feedback (i.e. increasing scores) below:**

[Reviewer eqVi] (Increased rating from 2 to 4 and then to 6)
+ **raised the score from 2 to 4 (at 24 Nov 2025, 10:29 AOE Time)**
    + Q6, Q8, Q2, W3: We explained the challenges of temporal QA. We also emphasized that our primary goal is to provide a comprehensive analysis of how different information sources and training techniques affect LLMs’ performance on the temporal reasoning with abstention task. We clarified the differences between our work and prior work of abstention with RL, explaining that our work is the first study of LLM abstention in Temporal QA.
     + Q1: We trained a classifier using Qwen2.5-1.5B-Instruct, with a classification head to classify whether the question is answerable. We presented the results of this experiment in the rebuttal.
     + Q3, Q4, Q5, Q7, Q9, W2: We explained the details of the composition of our SFT data, and the prompts used for frontier model evaluations. We elaborated the potential reasons for the differences between the results of SFT and RL, the failure of OOD tasks, and the surprising results on TimeQA-Hard.

+ **raised the score from 4 to 6 (at 26 Nov 2025, 06:35 AOE Time)**
Based on reviewer eqVi’s further suggestions, we edited and updated the paper in the following ways:
    + W1: We modified the abstract, introduction, results, and conclusion to address the reviewer’s comments. Reviewers confirmed that the storyline was improved.
    + W1: We fixed the issue with misordered figures to make sure the paper reads fluently. We also broke down Table 1 into two tables (i.e., Table 1 and Table 2), to improve clarity.
    + We added all the new experiments and discussions to the paper.

[Reviewer Ght9] (Kept the rating 8)
+ Q2: We added a LoRA-SFT experiment to examine if LoRA training can reduce the overfitting problem.
+ Q1, Q3, Q4, Q5, Q6: We clarify the reasons why it's worth focusing on temporal questions. We also discussed the interaction between abstention skills and the standard post-training pipelines. For OOD tasks, we discussed whether our approach would generalize to other types of unanswerable questions.

[Reviewer aoEZ] **(Increased rating from 2 to 4)**
+ raised the score from 2 to 4 (at 26  Nov 2025, 08:39 AOE Time)
    + W3:  We claimed that the main contribution of our work lies not in new training algorithms, but in the systematic study of temporal abstention in LLMs. Our work is the first to provide a comprehensive analysis of how different types of supervision, including implicit (context, sub-context, KG) vs. explicit (CoT), affect abstention behavior.
    + W4: We evaluated two LRMs (o4-mini and Qwen3-4B-Thinking) and presented the results in the rebuttal and paper.
    + W1, W2: We explained the reasons why we used timeQA datasets to study the task of temporal reasoning with abstention. We also clarified that GPT-o1 is used only to generate the reasoning steps, not ground-truth labels or external knowledge, and thus does not introduce knowledge distillation, leakage, etc.
+ Following the reviewer’s score updates and new comments, we further updated the paper with all the details included in the PDF:
    + We revised our stated contributions according to the reviewer’s suggestions, clarifying that we are the first to systematically study and compare how different information types and training techniques affect temporal reasoning with abstention behavior in LLMs.
    + We presented our interesting findings and novel insights.
Reviewer aoEZ stated and we quote “ to the best of my knowledge, are the first ones to study existing methods for abstention in Temporal QA.” We were hoping to have one more round of interaction with the reviewer when the glitch incident happened.

---

### Meta-Review · Area_Chair_dmT8 · 2025-12-26

**Summary:**

This paper presents a systematic empirical study on the ability of Large Language Models (LLMs) to abstain from answering in temporal QA tasks. The authors evaluate a variety of methods, including implicit reasoning cues (e.g., Knowledge Graph extraction and time-relevant sub-context filtering), Supervised Fine-Tuning (SFT), and a Reinforcement Learning (RL) pipeline using the GRPO framework. Through extensive experiments—including several added during the rebuttal—the paper demonstrates that while standard prompting and implicit reasoning often fail—frequently inducing over-confidence rather than accuracy—RL paired with a Chain-of-Thought (CoT) SFT cold start effectively teaches the model when to abstain. Notably, a 1.5B model trained with this pipeline was shown to outperform GPT-4o on the TimeQA-Hard benchmark.

The submission addresses an important and timely problem. While the technical novelty of the individual components is limited, the systematic investigation of how different information types (implicit vs. explicit) affect temporal abstention represents a valuable empirical contribution. One significant limitation, however, is the model's brittleness on out-of-distribution (OOD) non-temporal tasks. For instance, True Positives for abstention on SQUAD-v2 dropped from 1266 in the base model to just 1 after RL training.

The authors' defense of these results could be interpreted as arguing that identifying this limitation itself constitutes a unique contribution. While I disagree with such an interpretation, I find the empirical findings valuable enough to warrant inclusion in the conference, provided the limitations are handled with more rigor. I recommend acceptance, but strongly urge the authors to include concrete error analysis in the final version (e.g., adding specific examples of failures on non-temporal QA datasets) to shed light on why the RL-tuned model suffers from such failures.

**Reviewer Concerns:**

Two reviewers (eqVi, aoEZ) initially criticized the lack of algorithmic novelty, noting that the RL pipeline and rule-based rewards follow standard paradigms. During the rebuttal, the authors successfully shifted the paper’s narrative to focus on its strength as a comprehensive empirical analysis of a complex reasoning domain. Most empirical concerns—including the addition of a classifier baseline and evaluations of recent Large Reasoning Models (LRMs) like o4-mini—were effectively addressed during the discussion period.

**Reviewer Scores:**

Two reviewers (eqVI, aoEZ) were negative before the rebuttal, mostly because of the initial framing of the paper, and minor concerns on the empirical analysis. During the rebuttal period, most of the empirical concerns were addressed with additional experiments and clarifications. Judging from the reviewer-author discussion comments, I believe two reviewers would have increased their scores.

The other reviewer (Ght9) remained positive from the initial review to the discussion phase.

---

### Decision · Program_Chairs · 2026-01-26

Accept (Poster)